

# Effects of two contrasting biochars on gaseous nitrogen emissions and intensity in intensive vegetable soils across mainland China

Changhua Fan, Hao Chen, Bo Li, Zhengqin Xiong*

Jiangsu Key Laboratory of Low Carbon Agriculture and GHGs Mitigation, College of Resources and Environmental

Sciences, Nanjing Agricultural University, Nanjing 210095, China

*Corresponding author (Z. Xiong): E-mail: zqxiong@njau.edu.cn;

Tel: +86-25-84395148; Fax: +86-25-84395210

**Abstract**

Biochar amendment to soil has been proposed as a strategy for sequestering carbon, mitigating climate change and enhancing crop productivity, but few studies have demonstrated the effects of different feedstock-derived biochars on the various gaseous nitrogen emissions (GNEs, $N_2O$, NO and $NH_3$) across the typical vegetable soils in China. A greenhouse pot experiment with five consecutive vegetable crops was conducted to investigate the effects of two contrasting biochar, namely, wheat straw biochar (Bw) and swine manure biochar (Bm) on GNEs, vegetable yield and gaseous nitrogen intensity (GNI) in four typical vegetable soils from the main vegetable production regions (Hunan province (HN), Shanxi province (SX), Shandong province (SD) and Heilongjiang province (HLJ)) that are representative of the intensive vegetable ecosystems across mainland China. Results showed that remarkable GNE mitigation induced by biochar occurred in SX and HLJ soils, whereas enhancement of yield occurred in SD and HLJ soils. Additionally, both biochars decreased GNI, with Bw mitigated $N_2O$ and NO emissions by 21.8–59.1 % and 37.0–49.5 % (except for SD), respectively, while Bm improved yield by 4.0–30.5 % (except for HN). Since the biochar's effects on the GNEs and vegetable yield strongly depended on the attributes of the soil and biochar, both soil type and biochar characteristics should be seriously considered before conducting large-scale application of biochar in order to achieve the maximum benefits under intensive greenhouse vegetable agriculture.

**Keyword:** Biochar, Intensive vegetable soil, Gaseous nitrogen emissions (GNEs), Gaseous nitrogen intensity (GNI)



## 1 Introduction

Agriculture accounted for an estimated emission of 4.1 (1.7–4.8) Tg N $yr^{-1}$ for $N_2O$ and 3.7 Tg N $yr^{-1}$ for NO, contributing 60 % and 10 %, respectively, to the total global anthropogenic emissions, largely due to increases of N fertilizer application in cropland (Ciais, 2013). The concentration of atmospheric $N_2O$, a powerful, long-lived, greenhouse gas, has increased from 270 parts per billion by volume (ppbv) in the pre-industrial era to ~ 324 ppbv (Ussiri and Lal, 2013); it has 298 times the global warming potential (GWP) of $CO_2$ on a 100-year horizon (IPCC, 2013) and also causes depletion of the ozone layer in the atmosphere (Ravishankara et al., 2009). In contrast, $NO_x$, which is mainly emitted as nitric oxide (NO), does not directly affect the earth's radiative balance but catalyzes the production of tropospheric ozone ($O_3$), which is a greenhouse gas associated with detrimental effects on human health (Anenberg et al., 2012) and crop production (Avnery et al., 2011). Additionally, along with the high nitrogen (N) application, ammonia volatilization is one of the major N loss pathways (Harrison and Webb, 2001) as well, with up to 90% coming from agricultural activities (Misselbrook et al., 2000; Boyer et al., 2002). As a natural component and a dominant atmospheric alkaline gas, $NH_3$ plays an important role in atmospheric chemistry and ambient aerosol formation (Langridge et al., 2012; Wang et al., 2015b). In addition to nutrient enrichment (eutrophication) of terrestrial and aquatic systems and global acidification of precipitation, $NH_3$ has also been shown to be a major factor in the formation of atmospheric particulate matter and secondary aerosols (Kim et al., 2006; Pinder et al., 2007), leading to potentially adverse effects on human and ecosystem health such as visibility degradation and threats to biodiversity (Powlson et al., 2008; Behera et al., 2013). Consequently, the release of various reactive N species results in lower N use efficiency in agricultural systems.

In China, vegetable production devotes an area of approximately $24.7 \times 10^6$ ha, equivalent to 12.4% of the total available cropping area, and the production represented 52 % of the world vegetable production in 2012 (FAO, 2015). Intensified vegetable cultivation in China is characterized by high N application rates, high cropping index and frequent farm practices. Annual nitrogen fertilizer inputs for intensively managed vegetable cultivation in rapidly developing areas are 3–6 times higher than in cereal grain cultivation in China (Ju et al., 2006; Diao et al., 2013; Wang et al., 2015a). As a result, great concern exists about excess N fertilizer application, leading to low use efficiency in intensive vegetable fields in China (Deng et al., 2013; Diao et al., 2013). Meanwhile, intensive vegetable agriculture is considered to be an important source of $N_2O$ (Xiong et al., 2006; Jia et al., 2012; Li et al., 2015b; Zhang et al., 2015) and NO production (Mei et al., 2009). Moreover, ammonia volatilization is another important N pathway in fertilized soil, resulting in large losses of soil-plant N (Pacholski et al., 2008; Zhang et al., 2011). Therefore, the reduction of reactive N loss becomes a central environmental challenge to meet the joint challenges of high production and acceptable environmental consequences in intensive vegetable production (Zhang et al., 2013).



Biochar is the dark-colored, carbon (C)-rich residue of pyrolysis or gasification of plant biomass under oxygen
($O_2$)-limited conditions, specifically produced for use as a soil amendment (Sohi, 2012). The amendment of agricultural
ecosystems with biochar has been proposed as an effective countermeasure for climate change (Smith, 2016). These
additions would increase soil carbon storage (Mukherjee and Zimmerman, 2013; Stavi and Lal, 2013), decrease GHG
emissions (Li et al., 2016), and improve soil fertility and crop production (Major et al., 2010; Liu et al., 2013). However,
some recent studies have reported no difference or even an increase in soil $N_2O$ emissions induced by biochar application
from different soils (Saarnio et al., 2013; Wang et al., 2015a). Still, $NH_3$ volatilization was enhanced by biochar
application in pasture soil (Clough et al., 2010), vegetable soil (Sun et al., 2014) and paddy soil in the wheat-growing
season (Zhao et al., 2014). Additionally, crop productivity responses to biochar amendments differed among various
biochars (Cayuela et al., 2014). These inconsistent results suggest that current biochar application to soil is not a
"one-size fit-all paradigm" because of the variation in the physical and chemical characteristics of the different biochars,
soil types and crop species (Field et al., 2013; Cayuela et al., 2014). Moreover, limited types of biochar (Spokas and
Reicosky, 2009) and soil (Sun et al., 2014) were involved in the experiments in previous studies. Thus, the evaluation of
the different types of biochar under the typical soils is imperative to gain a comprehensive understanding of potential
interactions before the large-scale application of biochars in intensive vegetable cropping system in China.
Therefore, a greenhouse pot experiment was conducted in an effort to investigate the effects of different types of
biochar on gaseous nitrogen emissions (GNEs), namely, $N_2O$, NO and $NH_3$, simultaneously in four typical intensified
vegetable soils across main vegetable production areas of mainland China. Overall, the objectives of this research were to
gain a comprehensive insight into the effects of the different types of biochar on the GNEs, vegetable yield and gaseous
nitrogen intensity (GNI) in intensively managed vegetable production in China.

## 2 Materials and methods


### 2.1. Experimental soil and biochar


Four typical greenhouse vegetable cultivation sites with a long history (more than 10 years) of conventional
cultivation were selected from Northeast, Northwest, Central and Eastern China (Fig. S1), namely, Phaeozem, Anthrosol,
Acrisol and Cambisol (FAO and ISRIC, 2012) from Jiamusi (46°48′ N, 130°12′ E), Heilongjiang province (HLJ);
Yangling (34°18′ N, 108°2′ E), Shanxi province (SX); Changsha (28°32′ N, 113°23′ E), Hunan province (HN) and
Shouguang (36°56′ N, 118°38′ E), Shandong province (SD), respectively were collected and represented a range of
differences in physicochemical properties and regions (Table S1). Soil samples were manually collected from the
cultivated layer (0–20 cm) after the local vegetable harvest in April, 2015. The samples were air-dried and passed through



a 5 mm stainless steel mesh sieve and homogenized thoroughly. Any visible roots and organic residues were removed
manually before being packed with the necessary amount of soil to achieve the initial field bulk density. Each pot
received 15 kg of 105 °C dry-weight-equivalent fresh soil. For the biochar amendment pots, sieved biochar (2 mm) was
mixed with the soil thoroughly before the experiment, equivalent to a 40 t ha$^{-1}$ dose (dry weight). No more biochar was
added later in the experimental period.

Two types of biochar, derived from two common agricultural wastes in China: wheat straw and swine manure,

hereafter referred to as Bw and Bm, respectively (Table S1). The Bw was produced at the Sanli New Energy Company in
Henan, China, by pyrolysis and thermal decomposition at 400–500 °C. The Bm was produced through thermal
decomposition at 400 °C by the State Key Laboratory of Soil Science and Sustainable Agricultural, Institute of Soil
Science, Chinese Academy of Sciences. In accordance with Lu (2000), the SOC was measured by wet digestion with
$H_2SO_4$–$K_2Cr_2O_7$, TN was determined by semi-micro Kjeldahl digestion, and soil texture was determined with the pipette
method. The soil pH and biochar pH were measured in deionized water at a volume ratio of 1:2.5 (soil to water) with a
PHS-3C mv/pH detector (Shanghai Kangyi Inc. China). The soil $NO_3^-$–N and $NH_4^+$–N were measured following the
two-wavelength ultraviolet spectrometry and indophenol blue methods, respectively, using an ultraviolet
spectrophotometer (HITACHI, UV-2900, Tokyo, Japan). Electric conductivity (EC) was measured by using a
Mettler-Toledo instrument (FE30-K, Shanghai, China) at a 1:5 (w:v) soil to water ratio. Cation exchange capacity (CEC)
was determined using the $CH_3COONH_4$ method. Dissolved organic carbon (DOC) was extracted from 5 g of the
biochar/soil with an addition of 50 ml deionized water and measured by a TOC analyzer (TOC-2000/3000, Metash
Instruments Co., LTD, Shanghai, China). Ash content was measured by heating the biochars at 750 °C for 4 h. The
specific surface area of the biochar material was tested using the Brunauer–Emmett–Teller (BET) method, from which
the N adsorption–desorption isotherms at 77 K were measured by an automated gas adsorption analyzer ASAP2000
(Micromeritics, Norcross, GA) with + 5% accuracy. Scanning electron microscopy (SEM) imaging analysis was
conducted using a HITACHI S-3000N scanning electron microscope.
*2.2. Experimental set-up and management*

The pot experiments were performed at the greenhouse experimental station of Nanjing Agricultural University,

China. Five vegetable crops were grown successively in the four vegetable soils during the experimental period. For each
type of soil, three treatments with three replicates were arranged in a completely random design: urea without biochar
(N), urea with wheat straw biochar (N+Bw), urea with swine manure biochar (N+Bm). In addition, phosphate and
potassium fertilizers in the form of calcium magnesium phosphate and potassium chloride, together with urea, were
broadcasted and mixed with soil thoroughly prior to sowing the vegetables. No topdressing events occurred because of



the frequent cultivation and short growth period for the leafy vegetables. Based on the vegetable growth, all pots received
equal amounts of water and no precipitation. Detailed information on the pot management practices is provided in Table
S2.
Each pot consists of a 30 cm × 30 cm (height × diameter) cylinder made of polyvinyl chloride (PVC). The top of
each pot was surrounded by a special water-filled trough collar, which allowed a chamber to sit on the pot and prevent
gas exchange during the gas-sampling period. Small holes (diameter of 1 cm) at the bottom of the pots were designed for
drainage. To prevent soil loss, a fine nylon mesh (< 0.5 mm) was attached to the base of the soil cores before packing.
*2.3. Measurement of $N_2O$, NO and $NH_3$*
The NO and $N_2O$ fluxes were measured simultaneously from each vegetable cultivation using a static opaque
chamber method (Zheng et al., 2008; Yao et al., 2009). A square PVC chamber of 35 cm × 35 cm × 40 cm (length ×
width × height) was temporarily mounted on the pot for gas flux measurement. The chamber was coated with sponge and
aluminum foil outside to prevent solar radiation heating the chamber. Gas samples for flux measurements were collected
between 8 and 10 a.m. on each measuring day to minimize the influence of diurnal temperature variation. Gas fluxes
were usually measured once a week and every other day for one week following fertilizer application. To measure the
$N_2O$ flux, four samples were collected from the headspace chamber using 20 ml polypropylene syringes at 0, 10, 20, and
30 min after chamber closure. The gas concentrations in the samples were analyzed within 12 h after sampling using an
Agilent 7890A gas chromatograph equipped with an electron capture detector (ECD) for $N_2O$ detection. The carrier gas
was argon-methane (50 %) at a flow rate of 40 ml min$^{-1}$. The column and ECD temperatures were maintained at 40 and
300 °C, respectively. The gas chromatography configurations described by Wang et al. (2013) were adopted for the gas
concentration analysis. $N_2O$ flux was calculated using the linear increases in gas concentration with time. Sample sets
were rejected unless they yielded a linear regression value of $R^2 > 0.90$.
For each NO flux measurement, gas samples were collected from the same chamber that was used for the $N_2O$ flux
measurements (Yao et al., 2009). Before closing the chamber, an approximately 1.0 L gas sample from the headspace of
each chamber was extracted into an evacuated sampling bag (Delin Gas Packing Co., LTD, Dalian, China), and this
measurement was regarded as time 0 min for NO analysis. After 30 min under chamber enclosure conditions (i.e., after
the $N_2O$ sample collections were completed), another headspace gas sample with the same volume was extracted from
each chamber into another evacuated bag. Within 1 h after sampling, NO concentrations were analyzed by a model 42*i*
chemiluminescence NO–NO–$NO_X$ analyzer (Thermo Environmental Instruments Inc., Franklin, MA, USA). The NO
fluxes were derived from the concentration differences between the two collected samples. The NOx analyzer was
calibrated by a model 146*i* dynamic dilution calibrator system at the end of each crop-growing season.



The mean flux of $N_2O$ or NO during the experiment period was calculated as the average of all measured fluxes,
which were weighted by the interval between the two measurements (Xiong et al., 2006). The cumulative $N_2O$ was
calculated as the product of the mean flux and the entire duration.
The $NH_3$ volatilization was determined using the ventilation method (Zhao et al., 2010). The
phosphoglycerol-soaked sponge was replaced every day after each fertilization event for approximately one week. The
phosphoglycerol-soaked sponges used to collect the $NH_3$ samples were immediately extracted with 300 mL potassium
chloride (KCl) solution (1 mol $L^{-1}$) for 1 h. The concentration of ammonia nitrogen ($NH_4^+$–N) was measured using the
indophenol blue method at 625 nm (Sororzano, 1969) by ultraviolet spectrophotometry (HITACHI, UV-2900, Tokyo,
Japan, with 0.005 absorbance of photometric accuracy). The cumulative seasonal $NH_3$ volatilization was the sum of the
daily emissions during the measurement period.
*2.4. Auxiliary measurements*
Simultaneously with the determination of trace gas fluxes, the air temperature and the soil temperature at a depth of
5 cm were measured using thermally sensitive probes at each sampling date. Soil water content was also measured using
a portable water detector (Mode TZS-1K, Zhejiang Top Instrument Corporation Ltd., China) by the frequency domain
reflectometer   method at a depth of 5 cm. Measured soil water contents (v/v) were converted to water filled pore space
(WFPS) with the following equation:
WFPS = volumetric water content ($cm^3\ cm^{-3}$) / total soil porosity ($cm^3\ cm^{-3}$)                    (1)
Here, total soil porosity = [1 − (soil bulk density ($g\ cm^{-3}$) / 2.65)] with an assumed soil particle density of 2.65 ($g\ cm^{-3}$).
The total soil bulk density was determined with the cutting ring method according to Lu (2000).
After each vegetable crop reached physiological maturity, the fresh vegetable yield was measured by weighing the
whole aboveground and belowground biomass in each pot.
GNE = cumulative $N_2O$ + cumulative NO + cumulative $NH_3$ emissions (kg N $ha^{-1}$)                    (2)
GNI = GNE / vegetable fresh yield (kg N $t^{-1}$ yield)                    (3)
After the one-year pot experiment, a soil sample from each pot was blended carefully. One subsample was stored at
4 °C for determination of microbial biomass carbon (MBC), potential nitrification rate (PNR) and denitrification enzyme
activity (DEA) within 3 days. Another subsample was air-dried for analysis of SOC, TN, pH and EC. MBC was
determined by substrate-induced respiration using a gas chromatography (Anderson and Domsch 1978). PNR was
measured using the chlorate inhibition soil-slurry method as previously described (Kurola et al., 2005) with
modifications (Hu et al., 2016). DEA was quantified as described by Smith and Tiedje (1979).
*2.5. Data processing and statistics*



One-way ANOVA was performed to test the effects of the treatments on cumulative $N_2O$, NO and $NH_3$ emissions;
GNE; vegetable yield and GNI. Two-way ANOVA was used to analyze the effects of the biochar type; soil type; and their
interactions on $N_2O$, NO and $NH_3$ emissions, vegetable yield, GNE and GNI throughout the experimental period.
Multiple comparisons among the treatments were further explained using Tukey's HSD test. Significant differences were
considered at $P < 0.05$. All statistical analyses were performed using JMP ver. 7.0 (SAS Institute, Cary, NC, USA, 2007).
Pearson's correlation analysis was used to determine whether there were significant interrelationships between $N_2O$/NO
and PNR or DEA in each soil, using SPSS window version 18.0 (SPSS Inc., Chicago, USA).

**3. Results**
*3.1. Soil responses to biochar amendment*
Obvious differences in all observed soil properties existed among soil types (Table 1, $p < 0.001$), suggesting the
wide variations of soil characters across mainland China. Additionally, biochar amendments had significant influences on
all the soil properties ($p < 0.05$). Compared with N treatments, biochar amendments increased the SOC, TN and EC by
20.4–135.0 %, 0.5–21.2 % and 2.4–38.1 %, respectively, across all the soils. Compared with Bw, Bm amendment
resulted in higher contents of SOC and TN by 5.8–20.5 % and 9.5–14.2 %, respectively, whereas EC values were higher
by 3.3–21.5 % induced by Bw than Bm amendment over all soils. Additionally, biochar amendments significantly
enhanced soil pH by 0.27–0.64 and 0.08–0.10 units compared with N treatment in HN and SX soils ($p < 0.05$),
respectively, and higher values were detected with Bm than Bw amendment in all soils. Furthermore, biochar
amendments tended to increase MBC in SD and HLJ soils, and Bm performed better in MBC enhancements than Bw in
all soils.
As shown in Fig. 1, no consensus effects on PNR and DEA were observed with biochar amendments across all soils.
Compared with N treatment, biochar amendments significantly increased PNR in HLJ while exerted no influences on SD
soil (Fig. 1a). Compared with Bw, Bm amendment significantly increased PNR in HN and SX soils. Moreover, compared
with N, biochar amendments significantly reduced DEA by an average of 40.1 and 37.8 % in SX and HLJ ($p < 0.05$),
respectively, while producing no influence in SD soils (Fig. 1b). In comparison with Bw, remarkable enhancements were
observed by 42.5 and 74.4 % with Bm amendment in HN and SX soils, respectively ($p < 0.05$).
*3.2. Seasonal variations of $N_2O$ and NO emissions*
The dynamics of $N_2O$ fluxes from all N-applied treatments in the four vegetable soils were relatively consistent and
followed a sporadic and pulse-like pattern that was accompanied with fertilization, tillage and irrigation (Fig. 2). In
addition, peak $N_2O$ fluxes varied greatly. Most of the $N_2O$ emissions occurred during the Amaranth and Tung choy





growing periods, and there were several small emissions peaks during the Spinach and Coriander herb growing periods
due to lower N application rate (Table S2), soil temperature and water content (Fig. S2). The highest peaks of $N_2O$
emissions from HN, SX, SD and HLJ were 4133.7, 1784.0, 432.4 and 1777.2 $\mu g$ N $m^{-2} h^{-1}$, respectively. Although
biochar (Bw and Bm) application did not significantly alter the seasonal pattern of the $N_2O$ fluxes, they greatly lowered
some peaks of $N_2O$ emissions in the SX and HLJ vegetable soils (Fig. 2b and d).

Clearly, the NO fluxes demonstrated similar seasonal dynamics to the $N_2O$ fluxes (Fig. 3). Some relatively high

peak NO fluxes were still observed in the Spinach and Coriander herb planting seasons even though relatively low
temperatures occurred during these periods, primarily due to lower soil moisture which was suitable for NO production.
The NO fluxes ranged from -44.6 to 377.6 $\mu g$ N $m^{-2} h^{-1}$ across all soil types. Furthermore, some NO peaks were
significantly weakened with the Bw and Bm in the HN soil (Fig. 3a).
*3.3. Cumulative $N_2O$, NO and $NH_3$ emissions*

Cumulative $N_2O$ emissions varied greatly among soil types (Table 2, $p < 0.001$), from 1.97 to 31.56 kg N $ha^{-1}$ across

all the soils during the vegetable cultivation period (Table 3a). Biochar amendments had significant influences on the
cumulative $N_2O$ emissions, reducing $N_2O$ emissions by 13.7–41.6 % (Table 2). In comparison with the N treatment,
biochar amendment decreased $N_2O$ emissions by an average of 56.4 % and 47.5 % in SX and HLJ (Table 3a, $p < 0.05$),
respectively, with no remarkable influence in SD soil, indicating significant interactions between biochar and soil types
(Table 2, $p < 0.001$). Compared with Bm, Bw amendment performed better mitigation effects which decreased $N_2O$
emissions by 11.8–38.4 % across all the soils, significantly in HN soil (Table 3a, $p < 0.05$). In comparison with $N_2O$
emission, the cumulative NO emission was much smaller, with a remarkable variation of 0.20–8.99 kg N $ha^{-1}$ across all
soils (Table 3b). Though pronounced effects on NO emissions with a reduction by average of 45.8 % (Table 2, $p < 0.05$ ),
biochar amendments had no consensus effects across soils, reducing NO emissions in HN soil (Table 3b, $p < 0.05$) and
producing no remarkable influence on SD soil, which suggested significant interactions between biochar and soil types
(Table 2, $p < 0.001$). Compared with Bm, Bw amendment significantly reduced NO emissions in SX and HLJ soils
(Table 3b, $p < 0.05$). As shown in Table 4, $N_2O$ emissions had positive relationships with DEA both in SX and HLJ soils,
and were affected positively by PNR in HN soil. Additionally, NO emissions had positive correlations with both PNR
and DEA in SX soil. However, neither $N_2O$ nor NO emissions were influenced significantly by PNR and DEA in SD
soils.

As is shown in Table 3c, the cumulative $NH_3$ emissions fluctuated greatly from 4.72–7.57 kg N $ha^{-1}$across all the

soils. Though significantly enhancing $NH_3$ emissions (Table 2), biochar amendments produced no significant influences
on the $NH_3$ emissions relative to N treatment in most soils (Table 3c). A tendency was found for the cumulative $NH_3$



emissions in N+Bm to be higher than those in the N+Bw treatment, although this difference was not remarkable within
each soil. Additionally, stimulation effects were consistently present after the first fertilization event in each type of soil
(Fig. 4).
*3.4. Vegetable yield and gaseous N emissions intensity during the five-vegetable crop rotation*

The vegetable yields for the five consecutive vegetable crops are presented in Table 3e. Pronounced differences

existed among all soils (Table 2, $p < 0.001$). Biochar amendments exerted no significant effects on vegetable yield (Table
2). Compared with the N treatment, biochar amendments were prone to increase vegetable yield in SD and HLJ soils
against HN and SX soils (Tables 3e), denoting pronounced interactions between soil and biochar (Table 2, $p < 0.05$).
Compared with Bm, Bw amendment lowed total yield over all the soils (Table 3e), significantly in HN and SD soils ($p <$

0.05).

Table 3f presents the GNI during the whole experiment period, with a pronounced variation among soils (Table 2, $p$

$< 0.001$). The GNI was greatly affected by biochar amendment during the whole experiment period (Table 2, $p < 0.01$).
Compared to N treatment, biochar amendments reduced the GNI by 4.3–27.8 % across all soils, significantly in SX and
HLJ soils (Table 3f, $p < 0.05$). Moreover, there were no remarkable differences between Bw and Bm throughout all soils.

**4. Discussion**
*4.1. Biochar effects on GNEs across different soil types*

The effects of biochar amendment on the $N_2O$ and NO emissions may be positive, negative or neutral, largely

depending on the soil condition and the inherent characteristics of the biochar (Spokas and Reicosky, 2009; Nelissen et
al., 2014). In our study, effects of two biochars on the $N_2O$ and NO emissions did not follow a consensus trend across the
four typical vegetable soils (Table 3a, b). In agreement with Cayuela et al. (2014), who reported that the role of biochar in
mitigating $N_2O$ emission was maximal in soils close to neutrality, remarkable mitigation effects were observed in SX and
HLJ with the biochar amendments (Table 3a). These findings potentially resulted from the effects of the biochars on soil
aeration, C/N ratio and pH, which affected the N dynamics and N cycling processes (Zhang et al., 2010; Ameloot et al.,
2015). Moreover, mitigation of $N_2O$ emissions induced by biochar was probably due to the decreased denitrification in
SX and HLJ soils (Fig.1b and Table 4). In line with Obia et al. (2015), biochar decreased NO emissions in low-pH HN
soil (Table 3b), probably by inducing denitrification enzymes with higher activity, and then resulted in less NO
accumulation relative to $N_2$ production. Moreover, the liming effects of biochar prevented the chemical decomposition of
$NO_2^-$ to NO (Islam et al., 2008), leaving only enzymatically produced NO to accumulate. However, neither $N_2O$ nor NO
emission was significantly influenced by PNR or DEA, suggesting other processes might play vital roles in SD soil. In



addition, surplus N input in vegetable systems probably masked the beneficial effects of the biochar addition on the N
transformation (Wang et al., 2015a). Therefore, the underlying mechanism of how biochar affect those processes needs to
be illustrated in the further research.
On the other hand, different biochars may not produce universal influences on $N_2O$ emissions for the same soil due
to the distinct properties of the biochar (Spokas and Reicosky, 2009). In the current study, overall, in comparison with
Bm, the Bw amendment had more effective mitigation effects on $N_2O$ and NO emissions (Table 3a, b), largely due to the
following reasons. First, compared with Bw, the contents of the TN and DOC in Bm were 1.8- and 1.4-fold (Table S1),
respectively, which might supply extra N or C source for heterotrophic nitrification in the acidic HN soil, which made
Bm ineffective for reducing the $N_2O$ emissions (Table 3a). This result was in accordance with Li et al. (2015a), who
observed that biochar amendment had no significant influence on the cumulative $N_2O$ emissions, and even higher $N_2O$
emissions occurred when biochar was input. Additionally, as shown in Fig.1, Bm was more prone to stimulate PNR and
DEA, thus displaying lower mitigation ability than Bw. Second, compared with Bm, the C/N ratio was approximately
twofold in Bw (Table S1), presumably leading to more inorganic nitrogen being immobilized in biochar with a higher
C/N ratio (Ameloot et al., 2015), decreasing the available N for microorganisms. Last, as presented in Fig. S3 and Table
S1, Bw had more pores and surface area, having a better advantage over Bm in absorbing NO accordingly. Others have
found that the lower mitigation capacity of high-N biochars (e.g., manures or biosolids) is probably due to the increased
N release in the soil from the biochar (Schouten et al., 2012). To our knowledge, very few studies have investigated
biochar effects on NO emissions (Nelissen et al., 2014; Obia et al., 2015), and the mechanisms through which biochar
influence NO emissions are not elucidated yet. Therefore, more research is needed to clarify the underlying mechanisms
of biochar on NO emission.
Intensive managed soils receiving fertilizer such as urea or anhydrous $NH_3$ and ruminant urine patches are potential
hot spots for $NH_3$ formation, where the use of biochar is expected to retain $NH_3$–N in the soil system (Clough and
Condron, 2010). Actually, the effects of biochar amendments on $NH_3$ volatilization largely depend on soil characteristics,
biochar types and duration time. Soil texture is an important factor impacting $NH_3$ transfer and release. More clay
contents were present in the SX soil (Table S1), which was limited in large soil pores, thus, the addition of porous
biochar could enhance the soil aeration, promoting $NH_3$ volatilization (Sun et al., 2014). Additionally, it was worthy to
note that cumulative $NH_3$ emissions were slightly higher in soils with the Bm than those with the Bw amendment (Fig. 4
and Table 3c) and that difference could presumably be attributed to less surface area and the much higher pH of Bm (Fig.
S3 and Table S1), resulting in weak adsorption and great liming effects. Overall, compared with previous studies (Ro et
al., 2015; Mandal et al., 2016), no significant reductions were found in cumulative $NH_3$ volatilizations over the whole



observation period when biochar was added to current vegetable soils. In general, freshly produced biochar typically has
very low ability to absorb ammonium (Yao et al., 2012). Over time, biochar surfaces are oxidized and increase adsorption
(Wang et al., 2016). Moreover, the recorded increase in CEC by Cheng et al. (2006) indicated that biochars that are
sufficiently weathered over a period would increase their ability to retain cations such as $NH_4^+$–N. Further, relatively
long-term experiments are required to elucidate the mechanism and duration of effect.
*4.2. Biochar effects on vegetable yield and GNI across different soil types*

The application of biochar is usually intended to increase crop yields, and evidence suggests this may be successful

(Schulz et al., 2013; Li et al., 2016). Due to its liming effect, biochar helps to improve the supply of essential macro- and
micronutrients for plant growth (Chan and Xu, 2009; Major et al., 2010). Enhancement of vegetable yield with bicohar
amendment occurred in SD and HLJ soils (Table 3e). However, no promotion of yield was observed with biochar
amendments in HN and SX. This could be attributed to exacerbated soil salinity, which inhibited the uptake of nutrients
and water (Ju et al., 2006; Zhou et al., 2010) and the growth of the soil microorganisms (Setia et al., 2011), leading to
unsustainable greenhouse vegetable production. Compared with other biochar (Jia et al., 2012), the higher amounts of ash
in Bw and Bm may contain high salts causing soil salinity (Hussain et al., 2016). After the addition of the two salt-rich
biochars, the EC values of HN and SX vegetable soils increased and reached the limits to tolerance for the leafy
vegetables (Shannon and Grieve, 1998).

Additionally, the mixed performance of biochars as an amendment is related to the wide diversity of

physicochemical characteristics that translates into variable reactions in soil (Novak et al., 2014). First, compared to Bw,
more DOC content was in the Bm (Table S1), through which more nutrients may be directly introduced to the soil
(Rajkovich et al., 2012). In addition, besides their large amount of plant-available nutrients (Hass et al., 2012), manure
biochars have been generally considered significant for improving soil fertility by promoting soil structure development
(Joseph et al., 2010), with the result that Bm was found superior to Bw in vegetable production enhancement (Table 3e).
As biochar effects on vegetable yield were variable, both biochar properties and soil conditions and crop species ought to
be taken into account comprehensively before applying biochar to a certain soil condition.

Here, we assessed two feedstock-derived biochar effects on GNI in typical cultivated vegetable soils across

mainland China. Overall, biochar amendments reduced GNI over all the soils, with the magnitude largely depending on
soil type. Remarkable reduction in GNI had been detected due to the efficient mitigation induced by biochar in SX and
HLJ (Table 3f). However, despite enhanced vegetable yield, no significant decreases in GNI were observed in SD,
mainly because of the absence of mitigation effects on $N_2O$, NO and $NH_3$ emissions of biochars (Table 3a, b and c)
Additionally, mitigation efficacy on GNI were not notably different between Bw and Bm amendments across the four



soils, largely due to the divergent influences on GNE and yield that Bw was superior to Bm in mitigating the GNE while
Bm performed better in vegetable yield (Table 3d and e). Furthermore, from our perspective, economic
effectiveness/feasibility, such as the net ecosystem economic budget, should be considered synchronously in intensive
vegetable production before large-scale biochar application.

**5. Conclusion**
The study demonstrated that biochar amendments generally reduced $N_2O$ and NO emissions without influencing the
$NH_3$ emissions, while produced no consensus influences on yield though those effects were largely both biochar- and
soil-specific. Additionally, biochar amendments did decrease GNI in intensive vegetable soils across mainland China.
Furthermore, Bw was superior to Bm in mitigating the GNE whereas the Bm performed better in crop yield throughout
all soils. Consequently, both soil type and biochar characteristics need to be seriously considered before large-scale
biochar application under certain regions of intensive vegetable production.

**Acknowledge**
This work was jointly supported by the National Natural Science Foundation of China (41471192), Special Fund for
Agro-Scientific Research in the Public Interest (201503106) and the Ministry of Science and Technology
(2013BAD11B01).





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



**Table legends**
**Table 1**
Soil organic carbon (SOC), soil total nitrogen (TN), soil pH, electric conductivity (EC) and microbial biomass carbon
(MBC) as affected by different treatments across the four vegetable soils.

| Soil | Treatment | SOC (g kg$^{-1}$) | TN (g kg$^{-1}$) | pH | EC (ds m$^{-1}$) | MBC (mg kg$^{-1}$) |
|---|---|---|---|---|---|---|
| HN | N | 8.0±0.8c | 1.37±0.12b | 4.37±0.04c | 1.76±0.21b | 1353±119a |
| | N+Bw | 15.6±0.5b | 1.47±0.07b | 4.64±0.04b | 2.43±0.31a | 1173±49b |
| | N+Bm | 18.8±0.6a | 1.64±0.04a | 5.01±0.03a | 2.00±0.32ab | 1234±50ab |
| SX | N | 9.7±0.7c | 1.55±0.04b | 7.53±0.02b | 1.74±0.27b | 490±9a |
| | N+Bw | 15.6±0.8b | 1.62±0.06b | 7.61±0.05a | 2.25±0.22a | 495±16a |
| | N+Bm | 17.5±1.1a | 1.79±0.03a | 7.63±0.01a | 1.96±0.06ab | 504±18a |
| SD | N | 7.9±0.1b | 1.13±0.04b | 7.70±0.08a | 0.85±0.03b | 535±13b |
| | N+Bw | 14.2±0.6a | 1.20±0.04b | 7.66±0.03a | 0.92±0.04a | 554±10ab |
| | N+Bm | 15.5±1.4a | 1.37±0.06a | 7.71±0.03a | 0.87±0.02ab | 573±12a |
| HLJ | N | 29.9±0.5b | 2.19±0.04b | 6.91±0.05a | 0.83±0.03b | 921±44b |
| | N+Bw | 36.0±1.5a | 2.20±0.03b | 6.92±0.06a | 0.95±0.03a | 988±56b |
| | N+Bm | 38.1±1.8a | 2.41±0.01a | 6.94±0.04a | 0.92±0.06a | 1242±196a |
| ANOVA results | | | | | | |
| Biochar | | *** | *** | *** | *** | * |
| Soil | | *** | *** | *** | *** | *** |
| Biochar×Soil | | * | n.s. | *** | n.s. | ** |

Data shown are means ± standard deviations of three replicates. See Fig. 1 for treatments codes. Different letters within
the same column indicate significant differences among treatments within the same soil at $p < 0.05$ level.
***Significant at $p < 0.001$; **significant at $p < 0.01$; *significant at $p < 0.05$; n.s. not significant.

**Table 2**

Two-way ANOVA and mean effects of biochar (Bc) and soil (S) types on cumulative gaseous nitrogen ($N_2O$, NO and $NH_3$) emissions, gaseous nitrogen emission (GNE),

vegetable yield and gaseous nitrogen intensity (GNI) during the entire sampling period.

| Factors | DF | $N_2O$ emission | | | NO emission | | | $NH_3$ emission | | | GNE | | | Vegetable yield | | | GNI | | |
|---|---|---|---|---|---|---|---|---|---|---|---|---|---|---|---|---|---|---|---|
| | | SS | F | P | SS | F | P | SS | F | P | SS | F | P | SS | F | P | SS | F | P |
| Bc | 2 | 271.9 | 65.1 | *** | 46.4 | 174.7 | *** | 0.5 | 0.8 | n.s. | 380.5 | 86.4 | *** | 76.2 | 3.2 | n.s. | 0.1 | 7.9 | ** |
| S | 3 | 1429.9 | 228.1 | *** | 152.2 | 382.1 | *** | 4.1 | 3.8 | * | 2322.6 | 351.5 | *** | 4316.9 | 123.3 | *** | 2.3 | 110.3 | *** |
| Bc×S | 6 | 179.3 | 14.3 | *** | 33.4 | 41.9 | *** | 1.4 | 0.7 | n.s. | 234.5 | 17.7 | *** | 230.4 | 3.3 | * | 0.1 | 1.6 | n.s. |
| Model | 11 | 4009.7 | 174.5 | *** | 225.3 | 154.3 | *** | 29.1 | 7.5 | *** | 5290 | 218.3 | *** | 15962.0 | 124.4 | *** | 5.8 | 77.0 | *** |
| Error | 24 | 50.1 | | | 3.2 | | | 8.5 | | | 52.9 | | | 280.0 | | | 0.2 | | |
| biochar effect (n = 9) | | | | | | | | | | | | | | | | | | | |
| N mean | | 12.01±1.44a | | | 2.86±0.24a | | | 5.92±0.24b | | | 43.81±1.25b | | | 20.50±1.60a | | | 0.57±0.05a | | |
| N+Bw mean | | 7.01±0.58b | | | 1.55±0.14b | | | 6.65±0.27a | | | 43.53±1.67b | | | 14.94±0.84b | | | 0.45±0.04b | | |
| N+Bm mean | | 10.37±0.56a | | | 1.55±0.10b | | | 7.01±0.25a | | | 49.53±1.11a | | | 18.60±0.65a | | | 0.49±0.03ab | | |
| Soil effect (n = 9) | | | | | | | | | | | | | | | | | | | |
| HN mean | | 27.20±1.85a | | | 5.80±0.50a | | | 5.31±0.16c | | | 33.06±1.65c | | | 38.04±1.90a | | | 1.15±0.11a | | |
| SX mean | | 4.89±0.45b | | | 1.08±0.13b | | | 12.69±0.46a | | | 25.05±1.11d | | | 12.69±0.46b | | | 0.51±0.01b | | |
| SD mean | | 2.25±0.26c | | | 0.25±0.09c | | | 9.51±0.55b | | | 44.88±0.49b | | | 9.51±0.55c | | | 0.21±0.01c | | |
| HLJ mean | | 4.48±0.68b | | | 0.81±0.04b | | | 11.79±0.71a | | | 79.50±2.41a | | | 11.79±0.71b | | | 0.15±0.01c | | |

SS: the sum of squares.

F value: the ratio of mean squares of two independents samples.

P value: the index of differences between the control group and the experimental group. *, ** and *** indicate significance at $p < 0.05$, $p < 0.01$ and $p < 0.001$, respectively.

n.s. : not significant.

Data shown are means ± standard deviations of the nine replicates. See Fig. 1 for treatments codes. Different letters within the same column indicate significant differences

among treatments at $p < 0.05$ level.



**Table 3**
Cumulative gaseous nitrogen ($N_2O$, NO and $NH_3$) emissions, gaseous nitrogen emission (GNE), vegetable yield and
gaseous nitrogen intensity (GNI) under the different treatments across the four soils.

| Treatments | HN | SX | SD | HLJ |
|---|---|---|---|---|
| (a) Cumulative $N_2O$ emissions (kg N ha$^{-1}$) | | | | |
| N | 30.59±3.15a | 7.83±0.60a | 2.52±0.37a | 7.10±1.91a |
| N+Bw | 19.45±2.43b | 3.20±0.28b | 1.97±0.21a | 3.45±0.86b |
| N+Bm | 31.56±1.35a | 3.63±0.62b | 2.26±0.58a | 4.01±0.68b |
| (b) Cumulative NO emissions (kg N ha$^{-1}$) | | | | |
| N | 8.99±1.01a | 1.27±0.15a | 0.20±0.08a | 0.97±0.11a |
| N+Bw | 4.54±0.60b | 0.80±0.13b | 0.33±0.19a | 0.52±0.03b |
| N+Bm | 3.87±0.30b | 1.16±0.17a | 0.21±0.10a | 0.94±0.03a |
| (c) Cumulative $NH_3$ emissions (kg N ha$^{-1}$) | | | | |
| N | 4.72±0.27a | 5.79±0.54b | 6.34±0.51a | 5.67±0.42a |
| N+Bw | 5.09±0.38a | 6.83±0.74ab | 7.35±0.75a | 6.24±0.49a |
| N+Bm | 5.32±0.42a | 7.57±0.57a | 7.37±1.11a | 6.48±0.43a |
| (d) GNE (kg N ha$^{-1}$) | | | | |
| N | 44.30±3.13a | 14.89±1.33a | 9.06±0.80a | 13.74±1.67a |
| N+Bw | 29.08±2.21b | 10.82±1.14b | 9.64±0.88a | 10.21±0.92b |
| N+Bm | 40.76±1.66a | 12.36±0.74b | 9.84±0.49a | 11.42±0.27b |
| (e) Vegetable yield (t ha$^{-1}$) | | | | |
| N | 35.20±2.52a | 25.29±3.90a | 39.09±2.03b | 75.65±5.84b |
| N+Bw | 29.05±2.35b | 23.57±1.74a | 44.53±3.74b | 76.95±4.04ab |
| N+Bm | 34.93±2.87a | 26.30±2.63a | 51.00±3.18a | 85.89±3.29a |
| (f) GNI (kg N t$^{-1}$ yield) | | | | |
| N | 1.27±0.18a | 0.59±0.08a | 0.23±0.02a | 0.18±0.04a |
| N+Bw | 1.01±0.12a | 0.46±0.05b | 0.22±0.04a | 0.13±0.02b |
| N+Bm | 1.17±0.15a | 0.47±0.04b | 0.19±0.01a | 0.13±0.01b |

Data shown are means ± standard deviations of the three replicates. See Fig. 1 for treatments codes. Different letters
within the same column indicate significant differences among treatments within the same soil at $p < 0.05$ level.



**Table 4**

The correlations between $N_2O$ or NO emission and PNR or DEA in each soil.

| Item | HN | | SX | | SD | | HLJ | |
|------|------|------|------|------|------|------|------|------|
| | PNR | DEA | PNR | DEA | PNR | DEA | PNR | DEA |
| $N_2O$ | 0.75* | 0.66 | 0.49 | 0.76* | -0.10 | 0.16 | -0.82** | 0.70* |
| NO | 0.62 | -0.29 | 0.79* | 0.69* | -0.54 | 0.01 | -0.63 | 0.22 |

Asterisks indicated 0.05 level significances (*$p < 0.05$) and 0.01 level significances (**$p < 0.01$), n = 9.





**Figure legends**

**Fig. 1** Potential nitrification rate (PNR) and Denitrification enzyme activity (DEA) under different treatments in HN, SX, SD and HLJ soils. The three treatments with each soil were urea without biochar (N), urea with wheat straw biochar (N+Bw) and urea with swine manure biochar (N+Bm). Bars indicate standard deviation (mean + SD, n = 3). Different letters above the bars indicate significant differences among the different treatments within the same soil, at $p < 0.05$.

**Fig. 2** Temporal dynamics of soil $N_2O$ ($\mu$g N $m^{-2}h^{-1} \pm$ SD, n = 3) fluxes under different treatments in HN (a), SX (b), SD (c) and HLJ (d) vegetable soils with five consecutive vegetable crops. The solid arrows indicate fertilization. See Fig. 1 for treatments codes.

**Fig. 3** Temporal dynamics of soil NO ($\mu$g N $m^{-2}h^{-1} \pm$ SD, n = 3) fluxes under different treatments in HN (a), SX (b), SD (c) and HLJ (d) vegetable soils with five consecutive vegetable crops. The solid arrows indicate fertilization. See Fig. 1 for treatments codes.

**Fig. 4** Cumulative ammonia ($NH_3$) emissions from the HN (a), SX (b), SD (c) and HLJ (d) soils during the four nitrogen fertilization events F: every N fertilization event. The bars indicate the standard deviation of the mean (kg N $ha^{-1} \pm$ SD, n = 3) of each treatment for the sum of the four N fertilization events. See Fig. 1 for treatments codes. Different letters above the bars indicate significant differences among the different treatments for each soil, at $p < 0.05$.











