# Peer review of "Biochar reduces yield-scaled emissions of reactive nitrogen gases from"

_Biogeosciences, 2016_

## Referee Comment (RC1) · Anonymous Referee #1 · 16 Jan 2017

The manuscript tries to assess the combined effects of biochar application and soil types on N2O, NO, NH3 and crop productivity. The results can provide useful information, however, the language need some final check by a professional and the manuscript also suffers from some major and minor problems.

Major comments: 1. Many results confused me in this paper. i.e. the effect of N2O mitigation induced by biochar was probably due to the decreased DEA in SX and HLJ (fig.1b), it means the denitrification is the main process for the N2O production, however, the highest N2O emission occurred in HN with the lowest DEA(table 3), the result

is in contradiction? 2. Line 264, the authors suggested that N2O nor NO emissions were neither influenced by nitrification nor by denitrification, but by other process. Then what are the other processes? I think it should be more clearly discussed.

Specific comments: 1. The NH3 volatilization result affected by biochar and soil types is not mentioned in the abstract. 2. Line 19, "Bm improved yield...except for HN," but the increment in SX is also not significant. 3. Line 30, According to IPCC 2013, the global warming potential of N2O is 265 times of CO2 on a 100-year horizon. Please correct the data. Line 393-394, please modify. 4. Line 111, the experiment was conducted in the greenhouse experimental station, so how to use completely random design? 5. Line 255-257, could you maybe give some explanation for why a neutrality pH soil will cause mitigation effects of N2O emission? 6. Line 293-299, please only discuss significant effects. No significant reductions of NH3 volatilization were found in this study, NH3 volatilization increased after biochar applied though the effect did not significantly. So I think the discussion of how the biochar reduce NH3 volatilization is not necessary. And your interpretation of the results includes a lot of over speculations that cannot be logically derived from the results. 7. Line 304-310 and Line 311-318, should change place. 8. Line 324-326, this is a lengthy sentence that could be maybe divided into two parts. Please split the sentence between "Additionally...vegetable yield". 9. Line 326-328, the two sentences are dispensable. 10. Line 331-332, the conclusions of this study are either flawed. i.e. N2O and NO in SD show no significant changes among all treatments, and the conclusion cannot be drawn from your results only. Please modify. 11. Page 19-22, all the tables should be three-line tables. 12. Page 24-27, it is better to use the same y-axis scales in the same figure.

---

## Referee Comment (RC2) · Anonymous Referee #2 · 10 Feb 2017

The study "Effects of two contrasting biochars on gaseous nitrogen emissions and intensity in intensive vegetable soils across mainland China" is a relevant piece of research. It shows N2O, NO and NH3 emissions from a greenhouse experiment with 4 vegetable soils during 5 consecutive crops. Apart from the high value of the data itself, the results are interesting and open new research questions that the authors could follow in future works. The differences found in N2O mitigation in the different soils could be linked to different N2O formation pathways. Strong points: 1)It analyses several N gases. This is quite unique, since most studies just focus on N2O emissions. 2) It uses 4 types of soil (with contrasting properties) and it follows gas emissions for

a whole year with 5 crop rotations. Weak points: 3) Only 3 replicates are used. This is a bit limited for pot studies. A minimum of 5-6 replicates should be used. Of course using more replicates limits the number of treatments that can be included, but it would give statistically stronger results. 1) The writing could be improved. The language is mostly correct, but the story line is sometimes missing, making it hard to follow. There's a lot of "biochar increases in this treatment and this soil and it decreases in this other soil…." Please summarize and integrate results. This would make the paper much more attractive. It is not necessary to comment on all the results, they are shown in the figures and tables. 2) I do not totally agree with summing up NH3, NO and NO and naming it "gaseous N emissions". This misleads to think that these are all the N gas losses and the fact is that N2 emissions have not been contemplated in the study and could be substantial. Specific comments: The title could be improved. It should state the main results. For instance: "Biochar mostly decreases NO and N2O emissions but slightly increases NH3 emissions in intensive vegetable soils across mainland China". Or something similar. What is your main general conclusion? That should be your title. The abstract should also be better developed. For instance, it is not mentioned that wheat straw biochar performs better than the manure biochar regarding N2O mitigation. Line 89. Please also include the amount of biochar added to each pot, not only the Kg/Ha. Line 191: substitute "enhanced" for "increased". Line 259-260. Please do not link your N2O results with your DEA results. From Figure 1 we cannot know if biochar is decreasing total denitrified N or decreasing the N2O/N2 ratio. Line 303: There a spelling mistake (bicohar)

Biochars should be characterized for elemental analysis (Corg, N, H, O). This is important since the atomic ratio H:Corg has been found to be a relevant index for N2O mitigation. The X axis in Figures 2 and 3 must be wrong. They start in 1/15 and they finish in 1/15. Does Figure 1 (DEA) only report N2O? Why Is N2 not included?

---

## Author Comment (AC1) · 16 Feb 2017

Dear Reviewer #1: Thank you very much for your great support and critical comments. Those comments are all valuable and very helpful for revising and improving our paper, as well as further important guidance for our researches. We have made corrections which we hope to meet with approval. Please see the following point-by –point answers and the suplementary file of manuscript with tracking system for your further evaluation.

1. Thank you for your nice comments! The main reason is that N2O production and mitigation in different soil type was governed by different processes. It's applicable to

[Figure]

SX and HLJ but not HN soil. There were no significant relations between N2O emissions and DEA in HN soil (Table 4), which indicated that denitrification was not the main process for the N2O production. Many researchers had reported that some other processes such as heterotrophic nitrification (Zhu et al., 2011; Cai et al., 2010), nitrifier denitrification (Zhu et al., 2013) are the main pathways of N2O emissions especially in the soil with low pH, low carbon content and high mineral N content (Wrage et al., 2001), which greatly match the soil properties of the vegetable soil from HN. Thus, due to the complex potential pathways in HN soil, the lowest DEA activity might influence but not determine the magnitude of N2O emissions in HN soils. Cai, Y.J., Ding, W.X., Zhang, X.L., Yu, H.Y., Wang, L.F., 2010. Contribution of heterotrophic nitrification to nitrous oxide production in a long-term N-fertilized arable black soil. Communications in Soil Science and Plant Analysis 41, 2264-2278. Wrage, N., Velthof, G., Van Beusichem, M., Oenema, O., 2001. Role of nitrifier denitrification in the production of nitrous oxide. Soil Biology and Biochemistry 33, 1723-1732. Zhu, T., Zhang, J., Cai, Z., 2011. The contribution of nitrogen transformation processes to total N2O emissions from soils used for intensive vegetable cultivation. Plant and Soil 343, 313-327. Zhu, X., Burger, M., Doane, T.A., Horwath, W.R., 2013. Ammonia oxidation pathways and nitrifier denitrification are significant sources of N2O and NO under low oxygen availability. Proceedings of the National Academy of Sciences of the United States of America 110, 6328-6333. 2. Thank you for your comments! The other processes that related to the N2O or NO emissions might be nitrifier denitrification and heterotrophic nitrification. We discussed more about the other processes that related to the N2O or NO emissions on Page 13 line 14-18. Specific comments: 1. The NH3 volatilization result affected by biochar and soil types is not mentioned in the abstract. A: Thank you very much for your comments! Biochar amendments generally stimulated the NH3 emissions with greater enhancement from Bm than Bw. We added these results on Page 3 line 14-15. 2. Line 19, "Bm improved yield. . .except for HN," but the increment in SX is also not significant. A: Yes, you are right. Bm improved yield by 13.5–30.5% (except for HN and SX).We have revised it on Page 3 line 12. Thank you! 3. Line 30,

According to IPCC 2013, the global warming potential of N2O is 265 times of CO2 on a 100-year horizon. Please correct the data. Line 393-394, please modify. A: Thank you! We have corrected the data 298 by 265 on Page 4 line 6 and modified the corresponding citation on Page 19 line 19-20. 4. Line 111, the experiment was conducted in the greenhouse experimental station, so how to use completely random design? A: Sorry for the inappropriate descriptions! Before the trial, we labeled all the pots, and then distributed them by casting lots in the experiment region. We have also deleted the world "completely" to make it more appropriate on Page 7 line 8. 5. Line 255-257, could you maybe give some explanation for why a neutrality pH soil will cause mitigation effects of N2O emission? A: Thank you for your comments! As reported before, N2O is produced during several N2O production pathways and its release to the atmosphere is almost entirely controlled by microbial activities. Among all the pathways, denitrification has been approved to be a main process in upland fertilized soils (Cheng et al., 2015), especially in vegetable field (Qu et al., 2014). As was shown in Fig 1b, Biochar amendments significantly decreased DEA in neutrality pH soils (SX and HLJ), which cause mitigation effects of N2O emission. However, biochar did not reduce the N2O emissions in acid and alkaline soil. Soil pHs lower than 5 can adversely affect the activity of nitrous oxide reductase (Liu et al., 2010) and biochar application could not consistently alleviate the adverse effect of such acid pHs. Additionally, nitrification would be the main N2O production pathways, and biochar amendment stimulated nitrification which could increase the N2O emissions in alkaline soil (Sánchez-García et al., 2014). Therefore, biochar amendments might have promising mitigation effects through altering the DEA in neutrality pH soils, in which denitrification tends to be the main N2O pathway. Cheng, Y., Wang, J., Zhang, J. B., Müller, C., & Wang, S. Q. (2015). Mechanistic insights into the effects of n fertilizer application on N2O-emission pathways in acidic soil of a tea plantation. Plant and Soil, 389(1), 45-57. Liu, B., Mørkved, P.T., Frostegård, Å., Bakken, L.R., 2010. Denitrification gene pools, transcription and kinetics of NO, N2O and N2 production as affected by soil pH. Fems Microbiology Ecology 72, 407-417. Sánchezgarcía, M., Roig, A., Sanchezmonedero, M.A., Cayuela,

M.L., 2014. Biochar increases soil N2O emissions produced by nitrification-mediated pathways. Frontiers in Environmental Science 2, 25. Qu, Z., Wang, J., Almøy, T., & Bakken, L. R. (2014). Excessive use of nitrogen in chinese agriculture results in high N2O/(N2O+N2O) product ratio of denitrification, primarily due to acidification of the soils. Global Change Biology, 20(5), 1685–1698. 6. Line 293-299, please only discuss significant effects. No significant reductions of NH3 volatilization were found in this study, NH3 volatilization increased after biochar applied though the effect did not significantly. So I think the discussion of how the biochar reduce NH3 volatilization is not necessary. And your interpretation of the results includes a lot of over speculations that cannot be logically derived from the results. A: Yes, you are right! We deleted those speculations about the mitigation of NH3 emissions on Page 14 line 17. Thank you! 7. Line 304-310 and Line 311-318, should change place. A: Thank you! We have exchange lines on Page 14 line 22-30 and Page 15 line 1-6. 8. Line 324-326, this is a lengthy sentence that could be maybe divided into two parts. Please split the sentence between "Additionally. . .vegetable yield". A: Thank you! We revised the lengthy sentence on Page 15 line 12-14. 9. Line 326-328, the two sentences are dispensable. A: Thank you! We have deleted the two sentences on Page 15 line 14. 10. Line 331-332, the conclusions of this study are either flawed. i.e. N2O and NO in SD show no significant changes among all treatments, and the conclusion cannot be drawn from your results only. Please modify. A: Yes, sorry for the inconvenience! We have modified the descriptions that biochar amendments generally reduced N2O and NO emissions (except for SD soil) in conclusion on Page 16 line 2. 11. Page 19-22, all the tables should be three-line tables. A: We have revised all the tables on Page 24-27 and Page 3 in the supplementary material. Thank you! 12. Page 24-27, it is better to use the same y-axis scales in the same figure. A: We have revised the figures on Page 30-32.

Thank you once again for your great support and comments!

Sincerely yours, Zhengqin (on behalf of all authors)

**Biochar can decrease the gaseous reactive nitrogen intensity in**

**intensive vegetable soils across mainland China**

Changhua Fan, Hao Chen, Bo Li, Zhengqin Xiong*

Jiangsu Key Laboratory of Low Carbon Agriculture and GHGs Mitigation, College of Resources and Environmental

Sciences, Nanjing Agricultural University, Nanjing 210095, China

*Corresponding author (Z. Xiong): E-mail: zqxiong@njau.edu.cn;

Tel: +86-25-84395148; Fax: +86-25-84395210

**Fig. 1.** revised manuscript with figures

BGD
**Supplementary information**

**Fig. S1** Map showing the sampling sites in China.

**Fig. S2** Dynamics of water filled pore space (WFPS), air temperature and soil temperature during the vegetable cultivation period.

**Fig. S3** Scanning electron microscope (SEM) images of the biochars derived from Bw (a, b and c) and Bm (d, e and f).

Same magnification for a and d (×50), b and e (×400) and c and f (×2000).

[Figure]
**Fig. 2.** revised supplementary

[Figure]

[Figure]

**Biochar can decrease the gaseous reactive nitrogen intensity in**
**intensive vegetable soils across mainland China**

Changhua Fan, Hao Chen, Bo Li, Zhengqin Xiong*
Jiangsu Key Laboratory of Low Carbon Agriculture and GHGs Mitigation, College of Resources and Environmental
Sciences, Nanjing Agricultural University, Nanjing 210095, China
*Corresponding author (Z. Xiong): E-mail: zqxiong@njau.edu.cn;
Tel: +86-25-84395148; Fax: +86-25-84395210

**Fig. 3.** with tracking system

---

## Author Comment (AC2) · 16 Feb 2017

Dear Reviewer: Thank you very much for your great support and critical comments. Those comments are all valuable and very helpful for revising and improving our paper, as well as further important guidance for our researches. We have made corrections which we hope to meet with approval. Please see the following point-by –point answers. 1. Thanks for your nice comments! However, it is relatively hard to get general results on all the parameters as affected by biochar in all the vegetable soil, which is largely depending on soil and biochar type. Thus, we have tried our best to summarize

and integrated the results on Page 10 lines 16-17, Page 11 lines 7 and 12-13. 2. Yes, you are right. It is true that N2 emissions could be substantial and were not contemplated in the present study. Therefore, we defined the NH3, N2O and NO emissions as GNrE short for "gaseous reactive N emissions", which would be more accurate and appropriate. In addition, we have changed the "GNE" and "GNI" into "GNrE" and "GNrI", respectively, throughout the manuscript. Specific comments: 1. The title could be improved. It should state the main results. For instance: "Biochar mostly decreases NO and N2O emissions but slightly increases NH3 emissions in intensive vegetable soils across mainland China". Or something similar. What is your main general conclusion? That should be your title. A: Thank you very much for your nice comments! Actually, general conclusions could not be drawn easily for those parameters due to the complexity of soil and biochar properties. Besides, vegetable yield is also another important index we concerned about. Taking into all the parameters (NH3, N2O and NO emissions and yield) into account, GNrI would be a better indicator evaluating biochar effects on various soils. Therefore, the title "Biochar can decrease the gaseous reactive nitrogen intensity in intensive vegetable soils across mainland China" might be more appropriate for the paper. Thank you so much! 2. The abstract should also be better developed. For instance, it is not mentioned that wheat straw biochar performs better than the manure biochar regarding N2O mitigation. A: Yes, thank you! We have added the result that Bw performs better than Bm regarding N2O mitigation on Page 3 line 11. 3. Line 89. Please also include the amount of biochar added to each pot, not only the Kg/Ha. A: Thank you! We have added the amount of biochar "282.6 g pot-1" on Page 6 line 12. 4. Line 191: substitute "enhanced" for "increased". A: Thank you! We have substituted "enhanced" for "increased" on Page 10 line 9. 5. Line 259-260. Please do not link your N2O results with your DEA results. From Figure 1 we cannot know if biochar is decreasing total denitrified N or decreasing the N2O/N2 ratio. A: Yes, you are right! We have deleted the sentence on Page 13 line 10. Thank you! 6. Line 303: There a spelling mistake (bicohar). A: We are sorry for the inconvenience. We have corrected it on Page 14 line 21. Thank you! 7. Biochars should be characterized

for elemental analysis (Corg, N, H, O). This is important since the atomic ratio H:Corg has been found to be a relevant index for N2O mitigation. A: Thank you for your nice recommendation. We have added the elemental analysis data (Corg, N, H, O) on Page 3 in the supplementary material and the corresponding measuring methods on Page 6 lines 22-25 in the material and method section. 8. The X axis in Figures 2 and 3 must be wrong. They start in 1/15 and they finish in 1/15. A: We are sorry for the inconvenience. We have corrected Figures 2 and 3 on Pages 30 and 31. 9. Does Figure 1 (DEA) only report N2O? Why Is N2 not included? A: Yes, it does. Based on the method for DEA determination, acetylene (10%, v/v) was added to inhibit N2O reductase activity (Yoshinari et al., 1977). Therefore, DEA indicated N2O emissions from the processes " NO3−→NO2−→ NO→N2O" not the final step from "N2O→N2". The reported N2O emissions should include the potential N2 emissions. Thank you for your understanding! Yoshinari T, Hynes R, Knowles R. Acetylene inhibition of nitrous oxide reduction and measurement of denitrification and nitrogen fixation in soil[J]. Soil Biology & Biochemistry, 1977, 9(3):177-183.

Thank you once again for your great support and comments!

Sincerely yours, Zhengqin (on behalf of all authors) ~ Prof. Zhengqin Xiong, PhD College of Resources and Environmental Sciences Nanjing Agricultural University Weigang #1, Nanjing, 210095 PRC zqxiong@njau.edu.cn 86-13605188915 (cell) 86-25-84395148 (O) ORCID https://orcid.org/0000-0003-4743-7325 ~

Please also note the supplement to this comment:
http://www.biogeosciences-discuss.net/bg-2016-487/bg-2016-487-AC2-supplement.pdf

---

## Author Comment (AC3) · 16 Feb 2017

[revised manuscript text omitted]

---

## Referee Comment (RC3) · Anonymous Referee #3 · 20 Feb 2017

This article is well structured, well written and it seems that the experiment was well performed. Results are well described and discussed.

Some other articles on biochar and greenhouse gas emissions have put forward several hypotheses on the effect of biochar on greenhouse gas emissions. Some of them are discussed in the text, but all possible hypotheses could be discussed more systematically in the discussion.

I also would have used soil types instead of site names for the treatments. I think this is more relevant.

[Figure]

Some other small remarks

- Line 26 agriculture accountS

- Lines 78-line 83 please make shorter sentences. It also seems that the words 'sites' and 'soils' are mixed up.

- Line 86: what do you mean with 'initial' soil bulk density? Was that the bulk density of the site where soil was collected and does it mean that all treatments had different soil bulk densities?

- Line 90 add 'were used'

- Line 192. It is mentioned that values were higher for Bm than for Bw amendments but this is only significant for HN; ie the soil with the lowest initial pH

- Line 193-194: Bm performed only significantly better in HN, so it did not perform better in all soils.

- Line 208-209: they greatly lowered some peaks of N20 emissions: how many occasions, what reduction %, was it significant?

- Line 243: lowered

- Line 277: how is inorganic nitrogen being immobilized in biochar with higher C/N ratio? What is the presumed mechanism?

- Line 303: biochar is written wrong

- Line 306-307: how can soil microorganisms lead to unsustainable greenhouse vegetable production?

- Table 2. NH3-emissoins. BC is not a significant factor but letters are different for the biochar treatments. How is this possible?

---

## Author Comment (AC4) · 24 Feb 2017

Thank you very much for your critical comments and great support! We have already answered all the comments you suggested , and we hope this revised manuscript can fit with the acceptable standard for Biogeosciences. Please see the attached point-by-point answers with the marked-up manuscript version for your further evaluation. The corrections for the previous two referees' comments are also included and combined in the attached file). Sincerely yours, Zhengqin (on behalf of all authors)

Reviewer #3 (Remarks to the Author):

[Figure]

This article is well structured, well written and it seems that the experiment was well performed. Results are well described and discussed. Some other articles on biochar and greenhouse gas emissions have put forward several hypotheses on the effect of biochar on greenhouse gas emissions. Some of them are discussed in the text, but all possible hypotheses could be discussed more systematically in the discussion. I also would have used soil types instead of site names for the treatments. I think this is more relevant. A: Thank you very much for your great support and critical comments. Those comments are all valuable and very helpful for improving our manuscript, as well as further important guidance for our researches. We have made corrections which we hope to meet with approval. Please see the following point-by-point answers. According to your recommendation, several hypotheses were put forward that: 1) biochar amendment could affect GNrEs, vegetable yield and GNrI in vegetable soils across mainland China, 2) those influences would vary among biochar and soil types. We have added those hypotheses on Page 5 line 18-20. Thank you! Also, we have renamed the treatments with soil type instead of site names throughout the manuscript. Thank you so much for your suggestion!

Some other small remarks 1. Line 26 agriculture accountS A: Thank you! We have changed the "accounted" to "accounts" on Page 4 line 2. 2. Lines 78-line 83 please make shorter sentences. It also seems that the words 'sites' and 'soils' are mixed up. A: We have split the long sentence into shorter sentences on Page 6 lines 3-8. Thank you! 3. Line 86: what do you mean with 'initial' soil bulk density? Was that the bulk density of the site where soil was collected and does it mean that all treatments had different soil bulk densities? A: In order to simulate the biochar effects on different types of soil as much as possible, initial soil bulk density (same to the field condition) was set for the individual soil type. Thank you! 4. Line 90 add 'were used' A: We are sorry for the inconvenience. We have added the "were used" on Page 6 line 15. 5. Line 192. It is mentioned that values were higher for Bm than for Bw amendments but this is only significant for HN; ie the soil with the lowest initial pH A: Yes, biochar can increase soil pH at different degree, and this elevation is more significant for acidic soil than alkaline soil (Chintala et al., 2014). We have made some revision about the description on Page 10 line 10. Thank you! Chintala R, Schumacher T E, McDonald L M, et al. Phosphorus sorption and availability from biochars and soil/biochar mixtures[J]. Clean–Soil, Air, Water, 2014, 42(5): 626-634. 6. Line 193-194: Bm performed only significantly better in HN, so it did not perform better in all soils. A: Yes, you are right! Although the MBC was slightly higher in Bm amendment than that in Bw in all soils, this difference reached significant level only in Phaeozem. We have revised it on Page 10 line 11-12. Thank you! 7. Line 208-209: they greatly lowered some peaks of N2O emissions: how many occasions, what reduction %, was it significant? A: There were six and two times that biochar amendment significantly lowered peaks of N2O emissions in Anthrosol and Phaeozem by 8.7-74.4% and 23.6-73.6%, respectively. We have already added some description about the reduction on N2O peaks as affected by biochar on Page 10 line 27-28. Thank you! 8. Line 243: lowered A: We are sorry for the inconvenience. We have corrected it on Page 11 line 30. Thank you! 9. Line 277: how is inorganic nitrogen being immobilized in biochar with higher C/N ratio? What is the presumed mechanism? AïijŽAlthough biochar generally have high carbon contents, labile C fraction exist in biochar simultaneously. The assimilation of labile C by microorganism resulted in microbial demand for inorganic N in soil solution (DeLuca et al., 2006), which lead more N being immobilized. The Bw with higher C/N ratio possessed more labile C (such volatile matter, Bw vs Bm, 23.9% vs 16.3%, data not shown), which make Bw more suitable for N immobilization (Deenik et al., 2010). DeLuca T H, MacKenzie M D, Gundale M J, et al. Wildfire-produced charcoal directly influences nitrogen cycling in ponderosa pine forests[J]. Soil Science Society of America Journal, 2006, 70(2): 448-453. Deenik J L. Charcoal volatile matter content influences plant growth and soil nitrogen transformations.[J]. Soil Science Society of America Journal, 2010, 74(4):1259-1270. 10. Line 303: biochar is written wrong A: We are sorry for the inconvenience. We have corrected the spelling of biochar on Page 14 line 21. Thank you! 11. Line 306-307: how can soil microorganisms lead to unsustainable greenhouse vegetable production? A: That is a good question. As long as we know, soil microorganisms are critical to the maintenance of ecosystem service because of their contribution to soil structure; decomposition of organic matter; toxin removal; biogeochemical cycling of carbon, nitrogen, phosphorous, and sulphur; and suppresiveness of plant pathogens (Chaparro et al., 2012; Ferris and Tuomisto, 2015; Larkin, 2015). Additionally, soil salinity has become an essential issue in vegetable ecosystem (Shi et al., 2009; Han et al., 2014), which reduces microbial activity, microbial biomass and changes microbial community structure, and result in the unsustainable for greenhouse vegetable production. Chaparro J M, Sheflin A M, Manter D K, et al. Manipulating the soil microbiome to increase soil health and plant fertility[J]. Biology and Fertility of Soils, 2012, 48(5):489-499. Ferris H, Tuomisto H. Unearthing the role of biological diversity in soil health[J]. Soil Biology & Biochemistry, 2015, 85:101-109. Larkin R P. Soil health paradigms and implications for disease management.[J]. Annual Review of Phytopathology, 2015, 53(1):199. Shi W M, Yao J, Yan F. Vegetable cultivation under greenhouse conditions leads to rapid accumulation of nutrients, acidification and salinity of soils and groundwater contamination in South-Eastern China[J]. Nutrient Cycling in Agroecosystems, 2009, 83(1):73-84. Han J, Luo Y, Yang L, et al. Acidification and salinization of soils with different initial pH under greenhouse vegetable cultivation[J]. Journal of Soils & Sediments, 2014, 14(10):1683-1692. 12. Table 2. NH3-emissoins. BC is not a significant factor but letters are different for the biochar treatments. How is this possible? A: Sorry for the inconvenience! We have rechecked and revised the data on Table 2 on Page 25 and modified the corresponding description on Page 11 lines 21-22. Thank you so much!

Thank you very much once again for your helpful comments! Best Regards! Zhengqin ~

[Figure]

Please also note the supplement to this comment:
http://www.biogeosciences-discuss.net/bg-2016-487/bg-2016-487-AC4-
supplement.pdf
* * *
[Figure]

**Supplement:**

[revised manuscript text omitted]

| N+Bw mean | | 7.01±0.58b | | | 1.55±0.14b | | | 6.37±1.02a | | | 14.94±0.84b | | | 43.53±6.31a | | | 0.45±0.04b | | |
| N+Bm mean | | 10.37±0.56a | | | 1.55±0.10b | | | 6.68±1.10a | | | 18.60±0.65a | | | 49.53±6.91a | | | 0.49±0.03ab | | |
| Soil effect (n = 9) | | | | | | | | | | | | | | | | | | | |
| Acrisol mean | | 27.20±1.85a | | | 5.80±0.50a | | | 5.31±0.16c | | | 38.04±1.90a | | | 33.06±1.65c | | | 1.15±0.11a | | |
| Anthrosol mean | | 4.89±0.45b | | | 1.08±0.13b | | | 12.69±0.46a | | | 12.69±0.46b | | | 25.05±1.11d | | | 0.51±0.01b | | |
| Cambisol mean | | 2.25±0.26c | | | 0.25±0.09c | | | 9.51±0.55b | | | 9.51±0.55c | | | 44.88±0.49b | | | 0.21±0.01c | | |
| Phaeozem mean | | 4.48±0.68b | | | 0.81±0.04b | | | 11.79±0.71a | | | 11.79±0.71b | | | 79.50±2.41a | | | 0.15±0.
[revised manuscript text omitted]

| N+Bw mean | | 7.01±0.58b | | | 1.55±0.14b | | | 6.37±1.02a | | | 14.94±0.84b | | | 43.53±6.31a | | | 0.45±0.04b | | |
| N+Bm mean | | 10.37±0.56a | | | 1.55±0.10b | | | 6.68±1.10a | | | 18.60±0.65a | | | 49.53±6.91a | | | 0.49±0.03ab | | |
| Soil effect (n = 9) | | | | | | | | | | | | | | | | | | | |
| Acrisol mean | | 27.20±1.85a | | | 5.80±0.50a | | | 5.31±0.16c | | | 38.04±1.90a | | | 33.06±1.65c | | | 1.15±0.11a | | |
| Anthrosol mean | | 4.89±0.45b | | | 1.08±0.13b | | | 12.69±0.46a | | | 12.69±0.46b | | | 25.05±1.11d | | | 0.51±0.01b | | |
| Cambisol mean | | 2.25±0.26c | | | 0.25±0.09c | | | 9.51±0.55b | | | 9.51±0.55c | | | 44.88±0.49b | | | 0.21±0.01c | | |
| Phaeozem mean | | 4.48±0.68b | | | 0.81±0.04b | | | 11.79±0.71a | | | 11.79±0.71b | | | 79.50±2.41a | | | 0.15±0.01c | | |

SS: the sum of squares.

F value: the ratio of mean squares of two independents samples.

[revised manuscript text omitted]

---

## Author Response (AR1)

Dear Editors and Reviewers, thank you very much for your critical comments and great support! We have answered all the comments you suggested, and we hope this revised manuscript can fit with the acceptable standard for Biogeosciences. Please see the attached point-by-point answers with the marked-up manuscript version for your further evaluation.

Sincerely yours,

Zhengqin (on behalf of all authors)

Editor comments:

(Remarks to the Author):

Your manuscript has now been seen by three referees, and all of them find your work of interest. I fully agree. I also appreciate that you have made several steps to address the suggestions and comments by the reviewers. However, upon closer inspection I believe several changes are still required before this manuscript can be accepted for publication.

Thank you very much for your great support and critical comments. Those comments are all valuable and very helpful for revising and improving our paper, as well as important guidance for our further researches. We have made corrections which we hope to meet with approval. Please see the following point-by-point answers.

1. Please have a very close look at your calculations. Table 2 seems to contain several inconsistencies. For instance, averaged across biochar treatments, the average $N_2O$ fluxes show strong differences between soil types, with fairly low SD-values. If this is correct, then how can you explain the relatively small SD-values for the averages for the individual biochar treatments across soil types? After all, these are averages of numbers that show enormous spread. There might a part of the calculation that I am not aware of that can explain these results, but for now it seems that something might have gone wrong here. At the very least these calculations need a better explanation. Possibly calculations need to be redone. I have several other concerns about table 2 as well, please see the attached file for details.

   A: Thank you for your critical and constructive comment. You are right. It's not appropriate to analyze the mean effects of biochar among four different vegetable soils due to the strong differences between soil types. According to your suggestions, we have rechecked and revised our Table 2 to make it concise and clear. We deleted the mean effects of biochar and soil types and revised the related descriptions in the manuscript according to the Two-way analysis as a revised Table 2. According to the Two-way ANOVA analysis, both biochar and soil types and their interactions mostly had significant influences on these main parameters ($N_2O$, NO, $NH_3$, GNrE, Vegetable yield and GNrI) in our study. In addition, the multiple comparisons among the treatments and soil types for those main parameters were assessed in the revised Table 3. We hope that will be good to present the main results. Thank you very much for your comments.

2. Although the manuscript is generally easy to follow, it still contains quite a lot of grammatical errors. I made a few corrections in the attached file, but these are by no means complete. It might be an idea to involve a native speaker to check for language.

   A: Thank you! We asked once again the American Journal Expert for help to polish the manuscript.

3. On a minor note: I wonder if you can come up with a more informative title? "The effect of X on Y" is very common, but it doesn't tell potential readers what they want to know, and it doesn't invite them to find out either. Here's a possible suggestion (which you are of course free to ignore): "Biochar reduces yield-scaled emissions of reactive nitrogen gases from

vegetable soils across China"

A: Thanks for your nice suggestion. We have adopted this revised title on Page 1 lines 1-2.

**Table 2**

Two-way ANOVA for the effects of biochar (Bc) and soil (S) types on cumulative $N_2O$, NO and $NH_3$ emissions, gaseous reactive nitrogen emission (GNrE), vegetable yield and gaseous reactive nitrogen intensity (GNrI) during the entire sampling period.

| Factors | DF | $N_2O$ emission | | | NO emission | | | $NH_3$ emission | | | GNrE | | | Vegetable yield | | | GNrI | | |
|---|---|---|---|---|---|---|---|---|---|---|---|---|---|---|---|---|---|---|---|
| | | SS | F | P | SS | F | P | SS | F | P | SS | F | P | SS | F | P | SS | F | P |
| Bc | 2 | 271.9 | 65.1 | *** | 46.4 | 174.7 | *** | 0.5 | 0.8 | n.s. | 380.5 | 86.4 | *** | 76.2 | 3.2 | n.s. | 0.1 | 7.9 | ** |
| S | 3 | 1429.9 | 228.1 | *** | 152.2 | 382.1 | *** | 4.1 | 3.8 | * | 2322.6 | 351.5 | *** | 4316.9 | 123.3 | *** | 2.3 | 110.3 | *** |
| Bc×S | 6 | 179.3 | 14.3 | *** | 33.4 | 41.9 | *** | 1.4 | 0.7 | n.s. | 234.5 | 17.7 | *** | 230.4 | 3.3 | * | 0.1 | 1.6 | n.s. |
| Model | 11 | 4009.7 | 174.5 | *** | 225.3 | 154.3 | *** | 29.1 | 7.5 | *** | 5290 | 218.3 | *** | 15962.0 | 124.4 | *** | 5.8 | 77.0 | *** |
| Error | 24 | 50.1 | | | 3.2 | | | 8.5 | | | 52.9 | | | 280.0 | | | 0.2 | | |

SS: the sum of squares.

F value: the ratio of mean squares of two independents samples.

P value: the index of differences between the control group and the experimental group. *, ** and *** indicate significance at $p < 0.05$, $p < 0.01$ and $p < 0.001$, respectively.

n.s.: not significant.

**Table 3**

Cumulative gaseous nitrogen ($N_2O$, NO and $NH_3$) emissions, gaseous reactive nitrogen emission (GNrE), vegetable yield and gaseous reactive nitrogen intensity (GNrI) under the different treatments across the four soils.

| Treatments | Acrisol | Anthrosol | Cambisol | Phaeozem |
|---|---|---|---|---|
| (a) Cumulative $N_2O$ emissions (kg N ha$^{-1}$) | | | | |
| N | 30.59±3.15aA | 7.83±0.60aB | 2.52±0.37aC | 7.10±1.91aB |
| N+Bw | 19.45±2.43bA | 3.20±0.28bB | 1.97±0.21aB | 3.45±0.86bB |
| N+Bm | 31.56±1.35aA | 3.63±0.62bB | 2.26±0.58aB | 4.01±0.68bB |
| (b) Cumulative NO emissions (kg N ha$^{-1}$) | | | | |
| N | 8.99±1.01aA | 1.27±0.15aB | 0.20±0.08aC | 0.97±0.11aBC |
| N+Bw | 4.54±0.60bA | 0.80±0.13bB | 0.33±0.19aB | 0.52±0.03bB |
| N+Bm | 3.87±0.30bA | 1.16±0.17aB | 0.21±0.10aC | 0.94±0.03aB |
| (c) Cumulative $NH_3$ emissions (kg N ha$^{-1}$) | | | | |
| N | 4.72±0.27aB | 5.79±0.54bA | 6.34±0.51aA | 5.67±0.42aA |
| N+Bw | 5.09±0.38aB | 6.83±0.74abA | 7.35±0.75aA | 6.24±0.49aAB |
| N+Bm | 5.32±0.42aB | 7.57±0.57aA | 7.37±1.11aA | 6.48±0.43aAB |
| (d) GNrE (kg N ha$^{-1}$) | | | | |
| N | 44.30±3.13aA | 14.89±1.33aB | 9.06±0.80aC | 13.74±1.67aB |
| N+Bw | 29.08±2.21bA | 10.82±1.14bB | 9.64±0.88aB | 10.21±0.92bB |
| N+Bm | 40.76±1.66aA | 12.36±0.74bB | 9.84±0.49aC | 11.42±0.27bBC |
| (e) Vegetable yield (t ha$^{-1}$) | | | | |
| N | 35.20±2.52aB | 25.29±3.90aC | 39.09±2.03bB | 75.65±5.84bA |
| N+Bw | 29.05±2.35bC | 23.57±1.74aC | 44.53±3.74bB | 76.95±4.04abA |
| N+Bm | 34.93±2.87aC | 26.30±2.63aD | 51.00±3.18aB | 85.89±3.29aA |
| (f) GNrI (kg N t$^{-1}$ yield) | | | | |
| N | 1.27±0.18aA | 0.59±0.08aB | 0.23±0.02aC | 0.18±0.04aC |
| N+Bw | 1.01±0.12aA | 0.46±0.05bB | 0.22±0.04aC | 0.13±0.02bC |
| N+Bm | 1.17±0.15aA | 0.47±0.04bB | 0.19±0.01aC | 0.13±0.01bC |

Data shown are means ± standard deviations of the three replicates. See Fig. 1 for treatments codes. Different lowercase letters within the same column indicate significant differences among treatments within the same soil at $p < 0.05$ level. Different capital letters within the same row indicate significant differences among soil types within the same treatment at $p < 0.05$ level.

Specific revisions:
1. Thank you! We have changed the "mitigated" to "mitigating" on Page 3 line 13.
2. Thank you! We have split the lengthy sentence and revised it on Page 3 lines 14-17.
3. We have replaced "Additionally, …application" with "Finally," on page 4 line 10. Thank you!
4. We have deleted the sentence "Consequently, …systems" on Page 4 line 17. Thank you!

5. We have modified the description that "Therefore, the reduction of reactive N loss is key to meet the joint challenges…" on Page 4 line 28. Thank you!

6. We have revised "would" by "have been suggested to" on Page 5 line 3. Thank you!

7. We have revised "from different soils" by "for various soils" on Page 5 line 6. Thank you!

8. We have substituted "Still" with "Besides" on Page 5 line 6. Thank you!

9. We have revised "system" by "systems" on Page 5 line 14. Thank you!

10. We have revised "intensified" by "intensively cropped" on Page 5 lines 16-17. Thank you!

11. According to your suggestion, we have split long sentence into several shorter and easier sentences on Page 6 lines 4-7. Thanks for your suggestion.

12. We have added verbs for the sentence on Page 6 line 15. Thank you!

13. We have changed "further explained" into "assessed" on Page 9 line 14. Thank you!

14. We have revised the description that Bm amendment increased SOC and TN by 5.8-20.5% and 9.5-14.2% ($p < 0.05$) on Page 10 line 7. Thank you!

15. We have revised the description that Bm increased MBC relative to Bw in all soils to make it clearer on Page 10 line 11. Thank you!

16. We have deleted "remarkable" on Page 10 line 16. Thank you!

17. We have revised the description that "with an average reduction of 45.8%" on Page 11 line 12. Thank you!

18. We have revised the description that "biochar effects differed between soils" on Page 11 line 12. Thank you!

19. Sorry for the inconvenience! We have rechecked and revised the data on Table 2 on Page 25. Thank you!

20. According to the Two-way ANOVA in Table 2 on Page 25, we got that biochar factor (Bc), without the consideration of its species, did not have a significant influence on vegetable yield. We have improved the description on Page 11 line 25. Thank you!

21. We have replaced the "neutrality" with "pH neutral" on Page 13 line 7. Thank you!

22. We have modified the description "probably by stimulating denitrification enzyme activity" on Page 13 line 11. Thank you!

23. We have modified the description "the liming effects of biochar may have prevented the chemical decomposition…" on Page 13 line 12. Thank you!

24. Sorry for the inconvenience. Different from the rest soils, no one significant relation was found between $N_2O$/NO and PNR or DEA, Which indicated some other processes might occur in Cambisol. We have modified the description on Page 13 line 13. Thank you!

25. Thank you! We have improved the description that compared with Bw, Bm had more the contents of the TN and DOC by 80% and 40%, respectively" on Page 13 line 25. Thank you!

26. We have corrected "Intensive" by "Intensively" on Page 14 line 9. Thank you!

27. Sorry for the inconvenience. We have modified the description to make it clear on Page 15 lines 4-6. Thank you!

28. We have improved the date format in Figs 2 and 3 on Pages 30-31 and added the description of the inserted panels on Page 28 lines 8-9. Thank you!

Reviewer #1 (Remarks to the Author):

The manuscript tries to assess the combined effects of biochar application and soil types on $N_2O$, NO, $NH_3$ and crop productivity. The results can provide useful information, however, the language need some final check by a professional and the manuscript also suffers from some major and minor problems.

Major comments:

1. Many results confused me in this paper. i.e. the effect of $N_2O$ mitigation induced by biochar was probably due to the decreased DEA in SX and HLJ (fig.1b), it means the denitrification is the main process for the $N_2O$ production, however, the highest $N_2O$ emission occurred in HN with the lowest DEA (table 3), the result is in contradiction?

2. Line 264, the authors suggested that $N_2O$ nor NO emissions were neither influenced by nitrification nor by denitrification, but by other process. Then what are the other processes? I think it should be more clearly discussed.

A: 1.Thank you for your nice comments! The main reason is that $N_2O$ production and mitigation in different soil type was governed by different processes. It's applicable to SX and HLJ but not HN soil. There were no significant relations between $N_2O$ emissions and DEA in HN soil (Table 4), which indicated denitrification was not the main process for the $N_2O$ production. Many researchers had reported that some other processes such as heterotrophic nitrification (Zhu et al., 2011; Cai et al., 2010), nitrifier denitrification (Zhu et al., 2013) are the main pathways of $N_2O$ emissions especially in the soil with low pH, low carbon content and high mineral N content (Wrage et al., 2001), which greatly match the soil properties of the vegetable soil from HN. Thus, due to the complex potential pathways in HN soil, the lowest DEA activity might influence but not determine the magnitude of $N_2O$ emissions in HN soils.

Cai, Y.J., Ding, W.X., Zhang, X.L., Yu, H.Y., Wang, L.F., 2010. Contribution of heterotrophic nitrification to nitrous oxide production in a long-term N-fertilized arable black soil. Communications in Soil Science and Plant Analysis 41, 2264-2278.

Wrage, N., Velthof, G., Van Beusichem, M., Oenema, O., 2001. Role of nitrifier denitrification in the production of nitrous oxide. Soil Biology and Biochemistry 33, 1723-1732.

Zhu, T., Zhang, J., Cai, Z., 2011. The contribution of nitrogen transformation processes to total $N_2O$ emissions from soils used for intensive vegetable cultivation. Plant and Soil 343, 313-327.

Zhu, X., Burger, M., Doane, T.A., Horwath, W.R., 2013. Ammonia oxidation pathways and nitrifier denitrification are significant sources of $N_2O$ and NO under low oxygen availability. Proceedings of the National Academy of Sciences of the United States of America 110, 6328-6333.

2. Thank you for your comments! The other processes that related to the $N_2O$ or NO emissions might be nitrifier denitrification and heterotrophic nitrification. We discussed more about the other processes that related to the $N_2O$ or NO emissions on Page 13 line 14-18.

Specific comments:

1. The $NH_3$ volatilization result affected by biochar and soil types is not mentioned in the abstract.

A: Thank you! Biochar amendments stimulated the $NH_3$ emissions (highest in SX), and Bm resulted in slightly higher $NH_3$ emissions than Bw did in all types of soils. We added these results on Page 3 line 13-14.

2. Line 19, "Bm improved yield. . .except for HN," but the increment in SX is also not significant.

A: Yes, you are right. Bm improved yield by 13.5–30.5% (except for HN and SX).We have revised it on Page 3 line 13. Thank you!

3. Line 30, According to IPCC 2013, the global warming potential of $N_2O$ is 265 times of $CO_2$ on a 100-year horizon. Please correct the data. Line 393-394, please modify.

A: Thank you! We have corrected the data 298 by 265 on Page 4 line 6 and modified the corresponding citation on Page 19 line 19-20.

4. Line 111, the experiment was conducted in the greenhouse experimental station, so how to use completely random design?

A: Sorry for the inappropriate descriptions! Before the trial, we labeled all the pots, and then distributed them by casting lots in the experiment region. We have also deleted the world "completely" to make it more appropriate on Page 7 line 9.

5. Line 255-257, could you maybe give some explanation for why a neutrality pH soil will cause mitigation effects of $N_2O$ emission?

A: Thank you for your comments! As reported before, $N_2O$ is produced during several $N_2O$ production pathways and its release to the atmosphere is almost entirely controlled by microbial activities. Among all the pathways, denitrification has been approved to be a main process in upland fertilized soils (Cheng et al., 2015), especially in vegetable field (Qu et al., 2014). As was shown in Fig 1b, Biochar amendments significantly decreased DEA in neutrality pH soils (SX and HLJ), which cause mitigation effects of $N_2O$ emission. However, biochar did not reduce the $N_2O$ emissions in acid and alkaline soil. Soil pHs lower than 5 can adversely affect the activity of nitrous oxide reductase (Liu et al., 2010) and biochar application could not consistently alleviate the adverse effect of such acid pHs. Additionally, nitrification would be the main $N_2O$ production pathways, and biochar amendment stimulated nitrification which could increase the $N_2O$ emissions in alkaline soil (Sánchez-García et al., 2014). Therefore, biochar amendments might have promising mitigation effects through altering the DEA in neutrality pH soils, in which denitrification tends to be the main $N_2O$ pathway.

Cheng, Y., Wang, J., Zhang, J. B., Müller, C., & Wang, S. Q. (2015). Mechanistic insights into the effects of n fertilizer application on $N_2O$-emission pathways in acidic soil of a tea plantation. Plant and Soil, 389(1), 45-57.

Liu, B., Mørkved, P.T., Frostegård, Å., Bakken, L.R., 2010. Denitrification gene pools, transcription and kinetics of NO, $N_2O$ and $N_2$ production as affected by soil pH. Fems Microbiology Ecology 72, 407-417.

Sánchezgarcá, M., Roig, A., Sanchezmonedero, M.A., Cayuela, M.L., 2014. Biochar increases soil $N_2O$ emissions produced by nitrification-mediated pathways. Frontiers in Environmental Science 2, 25.

Qu, Z., Wang, J., Almøy, T., & Bakken, L. R. (2014). Excessive use of nitrogen in chinese agriculture results in high $N_2O/(N_2O+N_2O)$ product ratio of denitrification, primarily due to acidification of the soils. Global Change Biology, 20(5), 1685–1698.

6. Line 293-299, please only discuss significant effects. No significant reductions of $NH_3$ volatilization were found in this study, $NH_3$ volatilization increased after biochar applied though the effect did not significantly. So I think the discussion of how the biochar reduce $NH_3$ volatilization is not necessary. And your interpretation of the results includes a lot of over speculations that cannot be logically derived from the results.

A: Yes, you are right! We had deleted some speculations about the mitigation of $NH_3$ emissions induced by biochar on Page 14 line 17. Thank you!

7. Line 304-310 and Line 311-318, should change place.

A: Thank you! We have exchange lines on Page 14 line 22-30 and Page 15 line 1-7.

8.  Line 324-326, this is a lengthy sentence that could be maybe divided into two parts. Please split the sentence between "Additionally. . .vegetable yield".

A: Thank you! We have split the lengthy sentence and revised it on Page 15 line 13-15.

9.  Line 326-328, the two sentences are dispensable.

A: Thank you! We have deleted the two sentences on Page 15 line 15.

10. Line 331-332, the conclusions of this study are either flawed. i.e. $N_2O$ and NO in SD show no significant changes among all treatments, and the conclusion cannot be drawn from your results only. Please modify.

A: Yes, sorry for the inconvenience! We have modified the descriptions that biochar amendments mostly reduced $N_2O$ and NO emissions in conclusion on Page 16 line 2.

11. Page 19-22, all the tables should be three-line tables.

A: We have revised all the tables on Page 24-27 and Page 3 in the supplementary material. Thank you!

12. Page 24-27, it is better to use the same y-axis scales in the same figure.

A: We have revised the figures on Page 30-32.

Reviewer #2 (Remarks to the Author):

The study "Effects of two contrasting biochars on gaseous nitrogen emissions and intensity in intensive vegetable soils across mainland China" is a relevant piece of research. It shows $N_2O$, NO and $NH_3$ emissions from a greenhouse experiment with 4 vegetable soils during 5 consecutive crops. Apart from the high value of the data itself, the results are interesting and open new research questions that the authors could follow in future works. The differences found in $N_2O$ mitigation in the different soils could be linked to different $N_2O$ formation pathways. Strong points: 1) It analyses several N gases. This is quite unique, since most studies just focus on $N_2O$ emissions. 2) It uses 4 types of soil (with contrasting properties) and it follows gas emissions for a whole year with 5 crop rotations. Weak points: 3) Only 3 replicates are used. This is a bit limited for pot studies. A minimum of 5-6 replicates should be used. Of course using more replicates limits the number of treatments that can be included, but it would give statistically stronger results.

1) The writing could be improved. The language is mostly correct, but the story line is sometimes missing, making it hard to follow. There's a lot of "biochar increases in this treatment and this soil and it decreases in this other soil …" Please summarize and integrate results. This would make the paper much more attractive. It is not necessary to comment on all the results, they are shown in the figures and tables.

2) I do not totally agree with summing up $NH_3$, $N_2O$ and NO and naming it "gaseous N emissions". This misleads to think that these are all the N gas losses and the fact is that $N_2$ emissions have not been contemplated in the study and could be substantial.

A:

1.  Thanks for your nice comments! However, it is relatively hard to get general results on all the parameters as affected by biochar in all the vegetable soil, which is largely depending on soil and biochar type. Thus, we have tried our best to summarize and integrated the results on Page 10 lines 16-17, Page 11 lines 7 and 12.

2.  Yes, you are right. It is true that $N_2$ emissions could be substantial and were not contemplated in the present study. Therefore, we defined the $NH_3$, $N_2O$ and NO emissions as GNrE short for

"gaseous reactive N emissions", which would be more accurate and appropriate. In addition, we have changed the "GNE" and "GNI" into "GNrE" and "GNrI", respectively, throughout the manuscript.

Specific comments:

1. The title could be improved. It should state the main results. For instance: "Biochar mostly decreases NO and $N_2O$ emissions but slightly increases $NH_3$ emissions in intensive vegetable soils across mainland China". Or something similar. What is your main general conclusion? That should be your title.

A: Thank you for your nice comment! Actually, general conclusions could not be drawn easily for all the parameters due to the complexity of soil and biochar properties. Besides, vegetable yield is also another important index we concerned about. Taking into all the parameters ($NH_3$, $N_2O$ and NO emissions and yield) into account, GNrI would be a better indicator evaluating biochar effects on various soils. Therefore, we revised the title as "Biochar can decrease the yield-scaled emissions of gaseous reactive nitrogen from vegetable soils across China". Thank you so much!

2. The abstract should also be better developed. For instance, it is not mentioned that wheat straw biochar performs better than the manure biochar regarding $N_2O$ mitigation.

A: Yes, thank you! We have added the result that Bw performs better than Bm regarding $N_2O$ mitigation on Page 3 lines 11-12.

3. Line 89. Please also include the amount of biochar added to each pot, not only the Kg/Ha.

A: Thank you! We have added the amount of biochar "282.6 g $pot^{-1}$" on Page 6 line 12.

4. Line 191: substitute "enhanced" for "increased".

A: Thank you! We have substituted "enhanced" for "increased" on Page 10 line 8.

5. Line 259-260. Please do not link your $N_2O$ results with your DEA results. From Figure 1 we cannot know if biochar is decreasing total denitrified N or decreasing the $N_2O/N_2$ ratio.

A: Yes, you are right! We have deleted the sentence on Page 13 line 10. Thank you!

6. Line 303: There a spelling mistake (bicohar).

A: We are sorry for the inconvenience. We have corrected it on Page 14 line 21. Thank you!

7. Biochars should be characterized for elemental analysis (Corg, N, H, O). This is important since the atomic ratio H:Corg has been found to be a relevant index for $N_2O$ mitigation.

A: Thank you for your nice recommendation. We have added the elemental analysis data (Corg, N, H, O) on Page 3 in the supplementary material and the corresponding measuring methods on Page 6 lines 22-25 in the materials and method section.

8. The X axis in Figures 2 and 3 must be wrong. They start in 1/15 and they finish in 1/15.

A: We are sorry for the inconvenience. We have corrected Figures 2 and 3 on Pages 30 and 31.

9. Does Figure 1 (DEA) only report $N_2O$? Why Is $N_2$ not included?

A: Yes, it does. Based on the method for DEA determination, acetylene (10%, v/v) was added to inhibit $N_2O$ reductase activity (Yoshinari et al., 1977). Therefore, DEA indicated $N_2O$ emissions from the processes " $NO_3^-{\rightarrow}NO_2^-{\rightarrow}$ $NO{\rightarrow}N_2O$" not the final step "$N_2O{\rightarrow}N_2$". The reported N2O emissions should include the potential $N_2$ emissions. Thank you!

Yoshinari T, Hynes R, Knowles R. Acetylene inhibition of nitrous oxide reduction and measurement of denitrification and nitrogen fixation in soil[J]. Soil Biology & Biochemistry, 1977, 9(3):177-183.

Reviewer #3 (Remarks to the Author):

This article is well structured, well written and it seems that the experiment was well performed. Results are well described and discussed.

Some other articles on biochar and greenhouse gas emissions have put forward several hypotheses on the effect of biochar on greenhouse gas emissions. Some of them are discussed in the text, but all possible hypotheses could be discussed more systematically in the discussion.

I also would have used soil types instead of site names for the treatments. I think this is more relevant.

A: According to your recommendation, several hypotheses were put forward that: 1) biochar amendment could affect GNrEs, vegetable yield and GNrI in vegetable soils across mainland China, 2) those influences would vary among biochar and soil types. We have added those hypotheses on Page 5 line 17-19. Thank you!

Also, we have renamed the treatments with soil type instead of site names throughout the whole manuscript. Thank you so much for your suggestion!

Some other small remarks

1. Line 26 agriculture accountS

    A: Thank you! We have changed the "accounted" to "accounts" on Page 4 line 2.

2. Lines 78-line 83 please make shorter sentences. It also seems that the words 'sites' and 'soils' are mixed up.

    A: We have split the long sentence into shorter sentences on Page 6 lines 4-7. Thank you!

3. Line 86: what do you mean with 'initial' soil bulk density? Was that the bulk density of the site where soil was collected and does it mean that all treatments had different soil bulk densities?

    A: In order to simulate the biochar effects on different types of soil as much as possible, initial soil bulk density (same to the field condition) was set for the individual soil type. Thank you!

4. Line 90 add 'were used'

    A: We are sorry for the inconvenience. We have added the "were used" on Page 6 line 15.

5. Line 192. It is mentioned that values were higher for Bm than for Bw amendments but this is only significant for HN; ie the soil with the lowest initial pH

    A: Yes, biochar can increase soil pH at different degree, and this elevation is more significant for acidic soil than alkaline soil (Chintala et al., 2014). We have made some revision about the description on Page 10 line 10. Thank you!

    Chintala R, Schumacher T E, McDonald L M, et al. Phosphorus sorption and availability from biochars and soil/biochar mixtures[J]. Clean–Soil, Air, Water, 2014, 42(5): 626-634.

6. Line 193-194: Bm performed only significantly better in HN, so it did not perform better in all soils.

    A: Yes, you are right! Although the MBC was slightly higher in Bm amendment than that in Bw in all soils, this difference reached significant level only in Phaeozem. We have revised it on Page 10 line 10-11. Thank you!

7. Line 208-209: they greatly lowered some peaks of $N_2O$ emissions: how many occasions, what reduction %, was it significant?

    A: There were six and two times that biochar amendment significantly lowered peaks of $N_2O$ emissions in Anthrosol and Phaeozem by 8.7-74.4% and 23.6-73.6%, respectively. We have already added some description about the reduction on $N_2O$ peaks as affected by biochar on

8. Line 243: lowered

A: We are sorry for the inconvenience. We have corrected it on Page 11 line 28. Thank you!

9. Line 277: how is inorganic nitrogen being immobilized in biochar with higher C/N ratio? What is the presumed mechanism?

A:Although biochar generally have high carbon contents, labile C fraction exist in biochar simultaneously. The assimilation of labile C by microorganism resulted in microbial demand for inorganic N in soil solution (DeLuca et al., 2006), which lead more N being immobilized. The Bw with higher C/N ratio possessed more labile C (such volatile matter, Bw vs Bm, 23.9% vs 16.3%, data not shown), which make Bw more suitable for N immobilization (Deenik et al., 2010).

DeLuca T H, MacKenzie M D, Gundale M J, et al. Wildfire-produced charcoal directly influences nitrogen cycling in ponderosa pine forests[J]. Soil Science Society of America Journal, 2006, 70(2): 448-453.

Deenik J L. Charcoal volatile matter content influences plant growth and soil nitrogen transformations.[J]. Soil Science Society of America Journal, 2010, 74(4):1259-1270.

10. Line 303: biochar is written wrong

A: We are sorry for the inconvenience. We have corrected the spelling of biochar on Page 14 line 21. Thank you!

11. Line 306-307: how can soil microorganisms lead to unsustainable greenhouse vegetable production?

A: That is a good question. As long as we know, soil microorganisms are critical to the maintenance of ecosystem service because of their contribution to soil structure; decomposition of organic matter; toxin removal; biogeochemical cycling of carbon, nitrogen, phosphorous, and sulphur; and suppresiveness of plant pathogens (Chaparro et al., 2012; Ferris and Tuomisto, 2015; Larkin, 2015). Additionally, soil salinity has become an essential issue in vegetable ecosystem (Shi et al., 2009; Han et al., 2014), which reduces microbial activity, microbial biomass and changes microbial community structure, and result in the unsustainable for greenhouse vegetable production.

Chaparro J M, Sheflin A M, Manter D K, et al. Manipulating the soil microbiome to increase soil health and plant fertility[J]. Biology and Fertility of Soils, 2012, 48(5):489-499.

Ferris H, Tuomisto H. Unearthing the role of biological diversity in soil health[J]. Soil Biology & Biochemistry, 2015, 85:101-109.

Larkin R P. Soil health paradigms and implications for disease management.[J]. Annual Review of Phytopathology, 2015, 53(1):199.

Shi W M, Yao J, Yan F. Vegetable cultivation under greenhouse conditions leads to rapid accumulation of nutrients, acidification and salinity of soils and groundwater contamination in South-Eastern China[J]. Nutrient Cycling in Agroecosystems, 2009, 83(1):73-84.

Han J, Luo Y, Yang L, et al. Acidification and salinization of soils with different initial pH under greenhouse vegetable cultivation[J]. Journal of Soils & Sediments, 2014, 14(10):1683-1692.

12. Table 2. $NH_3$-emissoins. BC is not a significant factor but letters are different for the biochar treatments. How is this possible?

A: Sorry for the inconvenience! We have rechecked and revised the data on Table 2 on Page

25 and modified the corresponding description on Page 11 lines 19-20. Thank you so much!

Thank you very much once again for your helpful comments!
Best Regards!
Zhengqin

Prof. Zhengqin Xiong, PhD
College of Resources and Environmental Sciences
Nanjing Agricultural University
Weigang #1, Nanjing, 210095 PRC
zqxiong@njau.edu.cn
86-13605188915 (cell)
86-25-84395148 (O)

**Biochar reduces yield-scaled emissions of reactive nitrogen gases from vegetable soils across China**

Changhua Fan, Hao Chen, Bo Li, Zhengqin Xiong*

Jiangsu Key Laboratory of Low Carbon Agriculture and GHGs Mitigation, College of Resources and Environmental

Sciences, Nanjing Agricultural University, Nanjing 210095, China

*Corresponding author (Z. Xiong): E-mail: zqxiong@njau.edu.cn;

Tel: +86-25-84395148; Fax: +86-25-84395210

**Highlights**

1. Two contrasting biochars affected Gaseous Nitrogen Intensity across 4 major vegetable soils in China.

2. Biochar affects gaseous Nr or yield largely depending on soil types.

3. Both biochars decreased GNrI with Bw mitigated mitigating gaseous Nr whereas Bm improved improving yield.

**Abstract**

Biochar amendment to soil has been proposed as a strategy for sequestering carbon, mitigating climate change and enhancing crop productivity, but few studies have demonstrated the general effects of different feedstock-derived biochars on the various gaseous reactive nitrogen emissions (GNrEs, $N_2O$, NO and $NH_3$) simultaneously across the typical vegetable soils in China. A greenhouse pot experiment with five consecutive vegetable crops was conducted to investigate the effects of two contrasting biochar, namely, wheat straw biochar (Bw) and swine manure biochar (Bm) on GNrEs, vegetable yield and gaseous reactive nitrogen intensity (GNrI) in four typical vegetable soils from the main vegetable production regions (Acrisol (Hunan province (AcrisolHN), Anthrosol (Shanxi province (AnthrosolSX), Cambisol (Shandong province (CambisolSD) and Phaeozem (Heilongjiang province (PhaeozemHLJ) which) that are representative of the intensive vegetable ecosystems across mainland China. Results showed that remarkable GNrE mitigation induced by biochar occurred in AnthrosolSX and PhaeozemHLJ soils, whereas enhancement of yield occurred in CambisolSD and PhaeozemHLJ soils. Additionally, both biochars decreased GNrI through reducing $N_2O$ and NO emissions by 36.4–59.1 % and 37.0–49.5 % for Bw (except for Cambisol), respectively, while through improving yield by 13.5–30.5 % for Bm (except for Acrisol and Anthrosol). with Bw performed better than Bm regarding $N_2O$ mitigation, with Bw mitigateding $N_2O$ and NO emissions by 21.8–59.1 % and 37.0–49.5 % (except for SD), respectively, while Bm improved yield by 4.013.5–30.5 % (except for HN and SX). Biochar amendments generally stimulated the $NH_3$ emissions with greater enhancement from Futhermore, Bm performed better than Bwregarding $N_2O$ mitigation by 11.8–38.4 % and Bm promoted yield better than Bw by 11.6–20.2%. We can infer that Biochar amendments stimulated the $NH_3$ emissions (highest in AnthrosolSX), and Bm resulted in slightly higher $NH_3$ emissions than Bw did in all types of soils. Since 
[revised manuscript text omitted]

---

## Editor Decision (ED1)

[revised manuscript text omitted]

---

## Author Response (AR2)

Associate Editor Decision: Publish subject to minor revisions (Editor review) (10 Apr 2017) by Kees Jan van Groenigen

Comments to the Author:

dear authors,

Thank you for this revised version of the manuscript. I believe that the comments by the reviewers have now mostly been addressed; the manuscript is almost ready to be accepted for publication. However, the manuscript still contains quite a few grammatical errors and awkward sentences. Please see the attached file for editorial suggestions.

We are so sorry that it must have taken much effort and time for you correcting these grammatical errors and awkward sentences throughout the manuscript. We appreciate your great patience and nice help! We are now incorporating all your corrections and comments into the revised version. Please see the following point-to-point answers with the marked-up manuscript version.

1. We have improved the first highlight on Page 2 line 2. Meanwhile, we improved the highlight 3 to avoid using acronyms in the highlight. We try to limit the character number to 85.

2. We have substituted "whereas" with "and" on Page 2 line 4. Several other occasions were corrected correspondingly.

3. GNrE was used instead of GNrEs throughout the manuscript representing all these gaseous reactive nitrogen emissions.

4. We also updated the soil names in supplementary Table S1.

5. We have improved the description "However, few studies have compared …" on Page 3 line 3. Thank you!

6. We have revised the description "in four vegetable soils…from Heilongjiang province" on Page 3 lines 7-9. Thank you!

7. We have revised the description "…respectively, and by…"on Page 3 line 12. Thank you!

8. We have revised "in" with "from" on Page 4 line 28. Thank you!

9. We have added "flux" on Page 8 line 13. Thank you!

10. We have put equation 2 at the very end of section 2.3 and added the sentence "Cumulative fluxes of $N_2O$, NO and $NH_3$ were added to calculate total gaseous reactive nitrogen gas emissions (GNrE):" on Page 8 lines 22-24. Thank you!

11. We have revised "suggesting" by "reflecting" on Page 10 line 3. Thank you!

12. We have revised "characters" by "characteristics" on Page 10 line 4. Thank you!

13. We have deleted the sentence "primarily…NO production" on Page 10 line 30. Thank you!

14. We have revised the "N+Bm" and "N+Bw" with "Bm" and "Bw" throughout the manuscript. Thank you!

15. We have revised the errorous description "consensus trend" with "consistent trend" throughout the manuscript. Thank you!

16. We have revised the description "…, which then resulted in" on Page 13 line 11. Thank you!

17. We have revised the description "However, different from the other soils….and denitrification might paly vital roles" on Page 13 lines 13-15. Thank you!

18. We have revised the description "future research needs… those processes" on Page 13 line 21. Thank you!

19. We have deleted "On the other hand" on Page 13 line 22. Thank you!

20. We have revised the description "First, compared to Bw, the contents of TN and DOC were 80% and 40% higher in Bm" on Page 13 line 25. Thank you!

21. We have revised the description "with biochar addition" on Page 13 line 29. Thank you!

22. We have added "higher" on Page 14 line 1. Thank you!

23. We have revised the description "Our results show that …and biochar types" on Page 14 lines 11-12. Thank you!

24. We have revised the description "High clay contents in the Anthrosol likely limited soil porosity" on Page 14 lines12-13. Thank you!

25. We have revised the description "…which was probably due to …the two biochars." on Page 14 lines 23-24. Thank you!

26. We have revised the description "We speculated that…" on Page 15 lines 1-2. Thank you!

27. We have deleted "leading to …vegetable production" on Page 15 line 3. Thank you!

28. We have deleted the sentence "However…yield enhancement (Table 3a, b and e)" on Page 15 line 10. Thank you!

29. We have inserted "." before "overall" on Page 15 line 10. Thank you!

30. We have deleted the sentence "while produced…biochar- and soil-specific" and added "…from four soils…across mainland China" on Page 16 lines 3-4. Thank you!

31. We have added the description "In contrast…both biochar- and soil-specific" on Page 16 lines 3-4. Thank you!

Thank you very much once again for your great patience and helpful comments!
Best Regards!

Zhengqin

Prof. Zhengqin Xiong, PhD

[revised manuscript text omitted]

0.05.

---

## Editor Decision (ED2)

[revised manuscript text omitted]

0.05.

---

## Author Response (AR3)

Associate Editor Decision: Publish subject to technical corrections (01 May 2017) by Kees Jan van Groenigen

Comments to the Author:

Dear authors,

Thank you for this revised version. The manuscript reads a lot better now, and it is almost ready to be published. I have a few very minor editorial suggestions, please see the manuscript for details. Finally, I noticed that a new meta-analysis on the effects of biochar on crop yields was published since your manuscript was submitted (Jeffery et al. 2017; Environmental Research Letters). While not strictly necessary, a quick reference to this paper would make the discussion more up-to-date.

Thank you very much for your great support and nice comments! We are now incorporating all your comments into the revised version to improve the manuscript. Please see the following point-to-point answers with the marked-up manuscript version.

1.  Improved the first highlight on Page 2 line 2.
2.  Revised "…nitrogen emissions (GNrE) of $N_2O$, NO and $NH_3$" on Page 3 line 4. Thank you!
3.  Corrected the word "biochars" on Page 3 line 6. Thank you!
4.  Removed those data "(1.7-4.8)" on Page 4 line 2. Thank you!
5.  Improved the description "…fertilization application and low N use efficiency…" on Page 4 line 23. Thank you!
6.  Added the sentence "GNrI was calculated …equation:" on Page 9 lines 5-6. Thank you!
7.  Improved the description "biochar amendments …in Phaeozem but had no effect…." on Page 10 line 13. Thank you!
8.  Corrected the word "emission" on Page 10 line 22. Thank you!
9.  Inserted "." at the end of the sentence on 10 Page line 30. Thank you!
10. Deleted "across …cultivation period" on Page 11 line 3. Thank you!
11. Improved the sentence "…N treatment, the effect of biochar amendment on $N_2O$ ….types (Table 3a)" on Page 11 line 5. Thank you!
12. Revised "in relation to" with "compared to" on Page 11 line 7. Thank you!
13. Deleted the word "However" and start a new paragraph on Page 13 line 14. Thank you!
14. Revised "where" with "and" on Page 14 line 11. Thank you!
15. Revised "in the soil ecosystem" with "under these conditions" on Page 14 line 11. Thank you!
16. Revised "were inconsistent" with "differed" on Page 14 line 24. Thank you!
17. Added the updated reference "Jeffery et al., 2017" on Page 5 line 4 and Page 14 lines 21 and 25 and the corresponding citation on Page 19 lines 23-24.
    Thank you very much once again for your patient support and helpful comments!

Best Regards!

Zhengqin

~~

Prof. Zhengqin Xiong, PhD

[revised manuscript text omitted]

0.05.

---

## Editor Decision (ED3)

[revised manuscript text omitted]